# Liquid water on cold exo-Earths via basal melting of ice sheets

**Lujendra Ojha** [1] ✉, **Bryce Troncone**[1], **Jacob Buffo** [2], **Baptiste Journaux**[3] & **George McDonald** [4]

Liquid water is a critical component of habitability. However, the production and stability of surficial liquid water can be challenging on planets outside the Habitable Zone and devoid of adequate greenhouse warming. On such cold, icy exo-Earths, basal melting of regional/global ice sheets by geothermal heat provides an alternative means of forming liquid water. Here, we model the thermophysical evolution of ice sheets to ascertain the geophysical conditions that allow liquid water to be produced and maintained at temperatures above the pressure-controlled freezing point of water ice on exo-Earths. We show that even with a modest, Moon-like geothermal heat flow, subglacial oceans of liquid water can form at the base of and within the ice sheets on exo-Earths. Furthermore, subglacial oceans may persist on exo-Earths for a prolonged period due to the billion-year half-lives of heat-producing elements responsible for geothermal heat. These subglacial oceans, often in contact with the planet's crust and shielded from the high energy radiation of their parent star by thick ice layers, may provide habitable conditions for an extended period.

The habitability potential of planets in the circumstellar habitable zone (CHZ) of M-dwarf stars are of great interest because M-dwarf stars make up 75% of the population of stars in the galaxy, and >40% of M-dwarfs stars are expected to harbor Earth-sized planets (exo-Earths) in their CHZ[1,2]. However, key differences between the stellar and planetary environments between M-dwarf stars and the more luminous Sun-like stars have led to a long-standing debate around the habitability of M-dwarf orbiting exo-Earths[3]. In particular, the higher fraction of stellar luminosity emitted in the X-rays and ultraviolet (UV) by M-dwarfs, relative to sun-like stars[4], as well as their exhibition of more flaring than sun-like stars[5], presents challenges for the surface habitability of these planets. High surface radiation levels, which can present significant challenges for the stability of biological molecules, can be expected from the tendency of these planets to be depleted in X-ray/UV shielding ozone[6], while flaring can lead to periodic, orders of magnitude higher surface X-ray fluxes[7]. Although such radiation only penetrates about 20 cm of liquid water[8], this radiation still poses a challenge for near-

surface aquatic environments where ocean circulation will periodically expose water from below this depth to the surface.

Even if the harsh effects of the M-dwarf stellar environment were absent, a significant fraction of the M-dwarf orbiting exo-Earths would still require substantial greenhouse warming for liquid water to be stable on the surface, given their relatively low equilibrium temperature ($T_{eq}$) (Fig. 1; Table 1). However, the efficacy of greenhouse warming depends on various factors such as albedo, cloud cover, greenhouse gas species, and their residence time in the atmosphere; parameters that are not well constrained for most Earth-sized exoplanets[9]. Another notable, common feature of these planets is tidal locking, possibly leading to an eyeball-like climate state, where most of the planet is frozen, with the exception of the substellar point, where liquid water may exist[10]. Over time, even the liquid water at the substellar point may completely freeze due to sea-ice drift, and the planet may resemble large icy moons of the solar system[10]. In such cold, icy, rocky planets, basal melting may provide an alternative

[1]Department of Earth and Planetary Sciences, Rutgers University, Piscataway, NJ, USA. [2]Thayer School of Engineering, Dartmouth College, Hanover, NH, USA. [3]Department of Earth and Space Science, University of Washington, Seattle, WA, USA. [4]Department of Earth Sciences, University of Oregon, Eugene, OR, USA. ✉ e-mail: luju.ojha@rutgers.edu

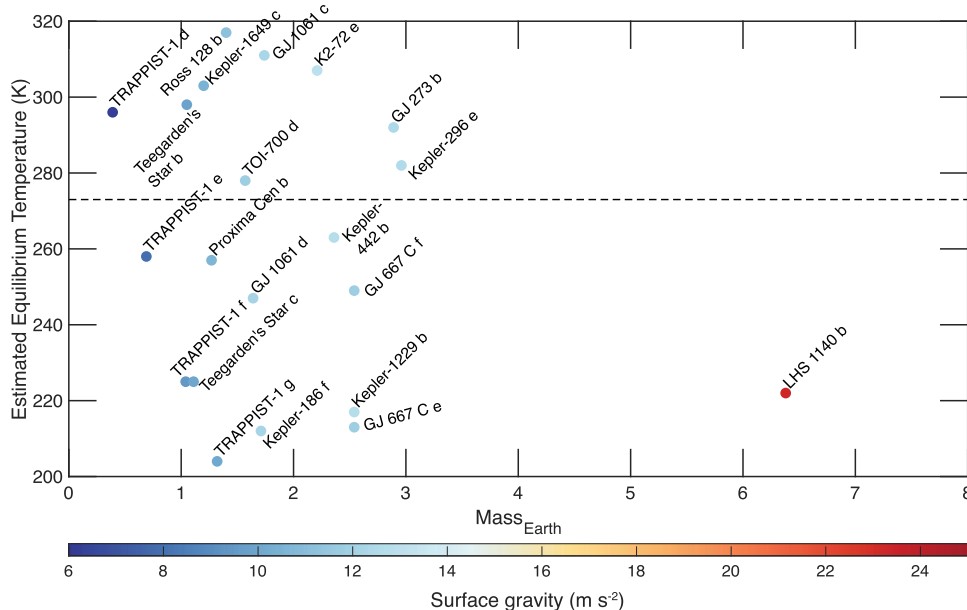

**Fig. 1 | Estimated equilibrium temperature ($T_{eq}$) and mass (relative to Earth) of various M-dwarf orbiting exo-Earths with a high Earth Similarity Index value.** The colors of the plot show the estimated surface gravity, and the dashed line marks the temperature of 273 K. The feasibility of basal melting on exoplanets with $T_{eq}$ less than 273 K is considered in this study. Source data are provided as a Source Data file.

**Table 1 | M-dwarf orbiting, potentially habitable planets with $T_{eq}$ < 273 K**

|           | Exoplanet         | $T_{eq}$ (K) | Surface gravity (m s$^{-2}$) | $M_{Earth}$ | $R_{Earth}$ | GEL of H$_2$O* (km) | Age of the star (Gyr)[Reference] |
|-----------|-------------------|--------------|------------------------------|-------------|-------------|---------------------|----------------------------------|
| *Reference* | *Earth*         | *255*        | *9.81*                       | *1*         | *1*         | *4.5*               | *4.6*                            |
| 1         | Proxima Cen b     | 257          | 10.6                         | 1.27        | 1.08        | 6.36                | $4.8^{+1.1}_{-1.4}$[79]          |
| 2         | GJ 1061 d         | 247          | 12.14                        | 1.64        | 1.15        | 7.24                | $7^{+0.58}_{-0.5}$[80]           |
| 3         | TRAPPIST-1 e      | 258          | 7.98                         | 0.69        | 0.92        | 4.76                | $7.6^{+2.2}_{-2.2}$[59]          |
| 4         | Kepler-442 b      | 263          | 12.68                        | 2.36        | 1.35        | 7.56                | $2.9^{+8.1}_{-0.2}$[81]          |
| 5         | GJ 667 C f        | 249          | 11.82                        | 2.54        | 1.45        | 7.05                | $3.5^{+1.5}_{-1.5}$[82]          |
| 6         | TRAPPIST-1 f      | 225          | 9.41                         | 1.04        | 1.04        | 5.61                | $7.6^{+2.2}_{-2.2}$[59]          |
| 7         | Teegarden's Star c | 225         | 10.04                        | 1.11        | 1.04        | 5.99                | $7^{+3}_{-3}$[83]                |
| 8         | Kepler-1229 b     | 217          | 12.68                        | 2.54        | 1.4         | 7.56                | $1.2^{+0.7}_{-0.3}$[84]          |
| 9         | Kepler-186 f      | 212          | 12.23                        | 1.71        | 1.17        | 7.29                | $4.0^{+0.6}_{-0.6}$[81]          |
| 10        | GJ 667 C e        | 213          | 11.82                        | 2.54        | 1.45        | 7.05                | $3.5^{+1.5}_{-1.5}$[82]          |
| 11        | TRAPPIST-1 g      | 204          | 10.12                        | 1.32        | 1.13        | 6.03                | $7.6^{+2.2}_{-2.2}$[59]          |
| 12        | LHS 1140 b        | 235          | 23.22                        | 6.38        | 1.64        | 13.85               | $5^{+2.4}_{-0}$[85]              |

*Global Equivalent Layer (GEL) of H2O assuming Earth like WMF 0.05%.

means of forming liquid water in a subsurface environment shielded from high-energy radiation (Fig. 2). In terrestrial glacial studies, the term basal melting is used to describe any situation where the local geothermal heat flux, as well as any frictional heat produced by glacial sliding, is sufficient to raise the temperature at the base of an ice sheet to its melting point[11]. Basal melting is responsible for the formation of subglacial liquid water lakes in various areas of Earth[12], such as the West Antarctic Ice Sheet[13], Greenland[14–16], and possibly the Canadian Arctic[17,18]. The Earth may have been globally glaciated at least three times[19–21]. During these periods, geothermal heat flow from the Earth's interior would have inhibited the deep ocean from completely freezing. As a result, only the shallow upper-extent of the ocean would have frozen[19]. Liquid water would have existed under the ice shell, and life could have survived through the snowball glaciations[22]. Similarly, basal melting of thick ice deposits during the Noachian era [>4 Ga] has been proposed as a potential solution to reconciling fluvial feature generation with the faint young sun paradox on Mars[23–26]. Although much debated[27–30], basal melting may also be responsible for the formation of a putative subglacial lake in the south pole of Mars today[31–35], where the mean annual surface temperature lies around 165 K. As such, basal melting may play an equally important role in the habitability of cold, icy exo-Earths (Fig. 2).

The feasibility of basal melting depends on the thermal and phase evolution of an ice sheet which is primarily governed by a planet's surface temperature and geothermal heat. Other factors, such as surface gravity and hydrosphere thickness, can also play a major role in the feasibility of basal melting, given the pressure-dependent melting temperature of water ice (Fig. 2). Notably, if the pressure in the hydrosphere is in excess of 0.2 GPa, dense high-pressure ice polymorphs (II, III, V, and VI) can form at the bottom trapping the interglacial ocean from access to vital solutes/nutrients limiting the interglacial ocean's habitability potential. Basal melting, if present, would enable water-rock interaction and the formation of buoyant aqueous solutions that can rise through the high-pressure ice layer to feed the interglacial ocean enabling potential life-sustaining conditions (Fig. 2).

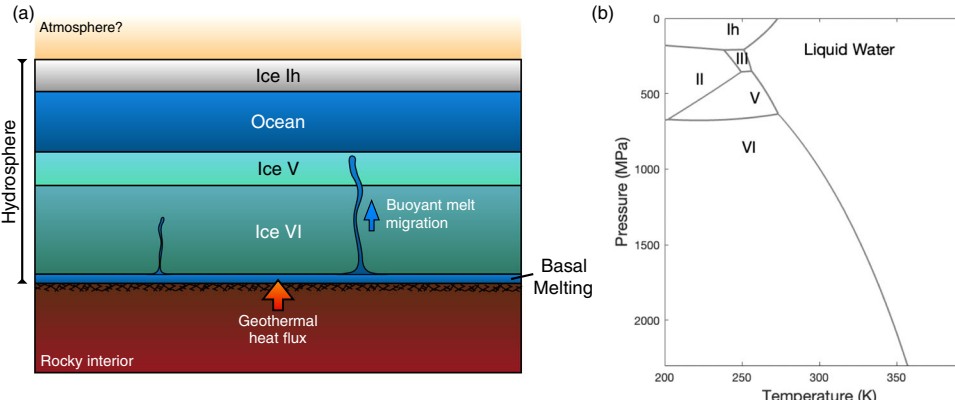

**Fig. 2 | Schematic of a basal melting model for icy exo-Earths. a** Due to the high surface gravity of super-Earths, ice sheets may undergo numerous phase transformations. Liquid water may form within the ice layers and at the base via basal melting with sufficient geothermal heat. If high-pressure ices are present, meltwater will be buoyant and migrate upward, feeding the main ocean. The red arrows show geothermal heat input from the planet's rocky interior. **b** Pure water phase diagram from the SeaFreeze representation[32] illustrating the variety of phases possible in a thick exo-Earth ice sheet. Density differences between the ice phases lead to a divergence from a linear relationship between pressure and ice-thickness.

Numerous M-dwarf orbiting exo-Earths with a high Earth Similarity Index (ESI) value, a multiparameter index that compares attributes such as mass, radius, and temperature of an exoplanet to Earth, have been discovered in the last decade[36,] (Fig. 1; Table 1). These exo-Earths' position within the CHZ of their parent stars also suggest that they could be water-rich planets. For example, the exo-Earth Proxima Centauri B orbits the star nearest to Earth. While the planet's water content is unknown, a wide range of internal structure models for Proxima Centauri B suggests that it could be a water-rich planet (up to 50% by mass)[37]. However, the stability of liquid water on the surface of Proxima Centauri B is hard to reconcile with the relatively low $T_{eq}$ of 257 K[38]. A recent study used an atmospheric general circulation model (GCM) coupled with a dynamic ocean GCM to show that even with an atmosphere consisting of 10,000 parts per million volume (ppmv) carbon dioxide ($CO_2$) and 2000 ppmv methane ($CH_4$), only a portion of the planet may be able to sustain liquid water[38]. Thus, if Proxima Centauri B is a water-rich planet, as its internal structure model suggests[37], a significant fraction of the water may exist in the form of solid ice, where basal melting may be feasible.

The internal structure models of several other exo-Earths suggest that they may be water-rich planets. For example, the internal structure model of the TRAPPIST-1 planetary systems suggest that they could potentially be more water-rich than Earth[39–41], although with notable uncertainty. Despite the strong far-UV photolysis of water ($H_2O$) and large $H_2$ escape rates expected from M-dwarf orbiting exo-Earths, models suggest that TRAPPIST-1 planets may have retained a significant amount of water[42]. While liquid water may be stable on the surface of TRAPPIST- 1 e with atmospheric $H_2O$ alone, the other two planets (TRAPPIST-1 f, g) require greenhouse gases such as $CO_2$ and a thick atmosphere to sustain surface liquid water[43]. LHS 1140 B is a super Earth with an estimated mass 6 times than of Earth's. A recent work[44] utilized the internal structure model developed by[45] to indicate that the water mass fraction (WMF) on LHS 1140 b could be as much as 4% (80 times more than Earth), leading to an average global ocean of $779 \pm 650$ km. That study's 1% confidence level corresponds to a WMF = 0.007, still about 1.5 times more water than Earth[44]. The transmission spectrum of LHS 1140 b from the Hubble Space Telescope also shows tentative evidence of water, but future observations are needed to confirm this putative detection[46]. The exo-Earth GJ 1061 d receives a similar amount of energy as Earth receives from the Sun and may be a water-rich planet as well[47]. GJ 667 C e and GJ 667 C f lie also within the CHZ of its host star and may be water-rich; however, they may still require a thicker atmosphere than Earth and greenhouse gases like $CO_2$ and $CH_4$ for the liquid water to be stable on their surface[48]. Kepler-442 b is a potentially water-rich exo-Earth with a high ESI value and potential for habitability[49]. Both Kepler 1229 b and 186 f have extremely low $T_{eq}$ and thus would require a thick atmosphere with green houses gases to sustain liquid water[50]. In the absence of a thick atmosphere, a significant fraction of water on these two exo-Earths may exist in the form of ice. While there are significant uncertainties regarding the possible presence and the volume of the hydrosphere on these planets, even with an Earth-like WMF of 0.05%, these bodies could contain kilometers-thick global ice sheets (Table 1), where basal melting may not only be possible but also play a significant role in habitability.

In this work, we model the thermophysical evolution of ice sheets of various thicknesses and demonstrate that basal melting is likely prevalent on M-dwarf orbiting exo-Earths, even with modest, Moon-like geothermal heat flow. We show that thick subglacial oceans of liquid water can form and persist at the base of and within the ice sheets on exo-Earths for a prolonged period. Our findings suggest that exo-Earths resembling the snowball Earth or the icy moons of Jupiter and Saturn may be common in the Milky-way galaxy.

## Results

We take a conservative approach and assume that the surface temperature ($T_s$) equals the estimated $T_{eq}$ (Fig. 1) for all exo-Earths considered in this study. Depth-dependent initial profiles of ice phases, density, specific heat, thermal conductivity, and melting temperature are estimated self-consistently (Fig. 3) and coupled with the thermal evolution model[23] to explore the feasibility of basal melting, while accounting for the time-varying ice phase evolution of the thick ice cap (see Methods subsection thermal evolution of ice sheets). An example of a thermal profile and the time-dependent phase evolution of a 2-km thick ice sheet on Proxima Centauri B, assuming a $T_s$ of 257 K and a basal heat flux of 30 mW m$^{-2}$, is shown in Fig. 3. In this scenario, basal melting occurs within a few hundred thousand years post-deposition of the ice resulting in approximately 800 m thick liquid water ocean at the ice-crust interface. The exo-Earths considered here have a wide range of surface temperature and gravity. Similarly, the hydrosphere, if present, on these bodies may encompass a wide range of thicknesses. Thus, we vary these parameters and run the thermophysical evolution models (Fig. 3) to ascertain their effect on the feasibility of basal melting.

## Effect of surface temperature, gravity, and ice-thickness on the feasibility of basal melting

The feasibility of basal melting is strongly dependent on $T_s$, with relatively low heat flow required for basal melting on planets with high $T_s$ (Fig. 4). For example, given the relatively high $T_s$ of Proxima Centauri B, TRAPPIST-1 e, and Kepler 442 b (260 K) (Table 1), even a 1-km thick ice sheet would undergo basal melting with a heat flow of 20–40 mW m$^{-2}$ (Fig. 4). In contrast, heat flow in excess of 175 mW m$^{-2}$ would be required for a 1-km ice sheet on Trappist 1 g ($T_s = 204$) to undergo basal melting (Fig. 4; Fig. 5a, b). Basal melting is also more likely to occur on planets with thicker ice sheets and higher surface gravity because the melting temperature of water-ice initially decreases with depth due to the pressure-reduced melting point of ice Ih (Fig. 2; Supplementary Fig. 1). Given the considerable uncertainty associated with the volume of the possible hydrosphere on these planets, we first consider a conservative scenario in which the WMF on exo-Earths is substantially lower than that of Earth and model the thermophysical evolution of 1–4 km thick icesheets (Fig. 4). Heat flow required for basal melting generally decreases with increasing ice thickness (Fig. 4). For example, while heat flow in excess of 30 mW m$^{-2}$ is needed for basal melting of 1-km thick ice on Kepler 442 b (Fig. 4), a 4-km ice sheet on the same planet would undergo basal melting with a modest Moon-like heat flow of 10 mW m$^{-2}$ (Fig. 4; Fig. 5c, d). If the exo-Earths considered here were to contain Earth-like WMF, the global equivalent layer (GEL) of hydrosphere on these bodies would lie between 5 and 14 km (Table 1). The general trend of the feasibility of basal melting with surface temperature, surface gravity, and ice thickness does not change when we consider thicker ice-sheets (Fig. 6); however, the total heat flow required for basal melting decreases notably (Fig. 6; Supplementary Fig. 2).

### Basal melting on super-Earths

If exo-Earths were to contain much thicker ice sheets, as proposed for LHS 1140 b, the basal pressure and the melting temperature could drastically increase. For example, at the base of a 75 km thick ice sheet on LHS 1140 b which has a surface gravity of 23 m s$^{-2}$, basal pressure would approach 2300 MPa, and basal temperatures

higher than 350 K would be required to generate meltwater at the bottom (Fig. 2). However, liquid water generated at the rock/high-pressure ice mantle boundary will be buoyant and can quickly migrate upward feeding the upper ocean below the ice Ih crust, as shown in previous study on high pressure ice mantle of icy moons[51,52] and in Fig. 2. Considering the lower end $T_s$ estimate of 235 K for LHS 1140 b[9], heat flow above 30 mW m$^{-2}$ can result in basal melting within 70 km ice sheets (Fig. 7). If the higher end $T_s$ estimate of 265 K is considered[9], heat flow of 5 mW m$^{-2}$ is sufficient for melting within the upper tens of km of the ice sheet (Fig. 7). Given the uncertainty in the water content of LHS 1140 b, we explore the feasibility of basal melting of ice sheets of a wide range of thicknesses (Fig. 7). On a super-Earth like LHS 1140 b, ice sheets in the 10–40 km thickness range are most likely to undergo basal melting due to the pressurized ice at the base, which has a reduced melting point (Fig. 7).

### Thickness and the stability of basal melt

All else being equal, the thickness of the basal melt increases with heat flow and time. For example, in Supplementary Fig. 3, we show the thermophysical evolution of a 10-km thick ice sheet on Proxima Centauri B for two different heat flow values (5 and 10 mW m$^{-2}$). In both cases, basal melting occurs within a few hundred thousand years post-deposition of the ice, and the thickness of basal melt increases until the system achieves thermal equilibrium. However, the basal melt resulting from a heat flow of 10 mW m$^{-2}$ is notably thicker than basal melt from a heat flow of 5 mW m$^{-2}$. Supplementary Fig. 4 shows similar results for GJ 1061 D. For computational efficiency and to demonstrate the feasibility of basal melting, we have limited the thermophysical evolution of ice for 5 million years. This time frame is adequate for the thermal equilibration of ice sheets and to ascertain whether a given heat flow is sufficient for basal melting. To assess the stability of basal melt over timescales relevant to the genesis of life on Earth (0.5–1 Gyr), we model the thermophysical evolution of a 1-km ice sheet on Proxima Centauri B for a billion years (Fig. 8). Over a billion years, geothermal heat flow can be expected to decline given the billion years half-life of most heat producing elements. Thus, we consider two scenarios: an

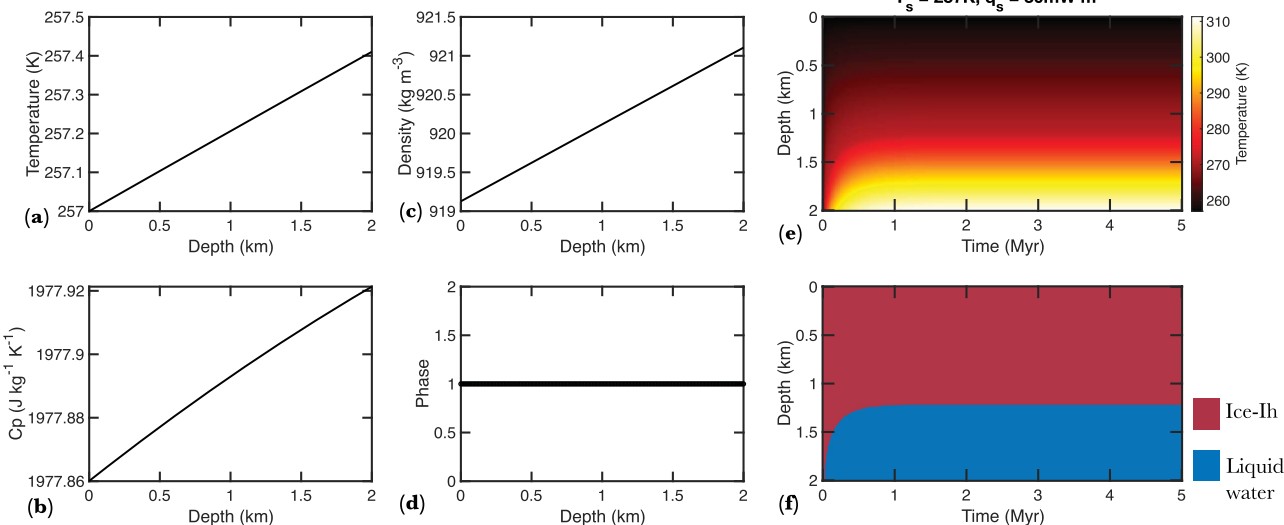

**Fig. 3 | Steady state initial profiles of various thermophysical parameters, temperature, and ice phases as a function of depth within a thick ice sheet on Proxima Centauri B assuming a surface temperature of 257 K. a** An example of initial steady-state profiles of temperature (**a**), specific heat (**b**), density (**c**), and ice-phase (**d**) as a function of depth in a 2 km thick ice sheet on exo-Earth Proxima Centauri B assuming a surface temperature of 257 K. The temperature is assumed to increase with depth in an adiabatic fashion. These initial profiles are coupled with a

thermal evolution model to explore the feasibility of basal melting and the time-varying ice phase evolution of the thick ice cap. **e** Temperature distribution as a function of depth and time on Proxima Centauri B assuming a 2 km thick ice sheet, $T_s$ of 257 K, and heat flow of 30 mW m$^{-2}$. **f** Ice phase evolution as a function of depth and time on Proxima Centauri B assuming a 2 km thick ice sheet, $T_s$ of 257 K, and heat flow of 30 mW m$^{-2}$.

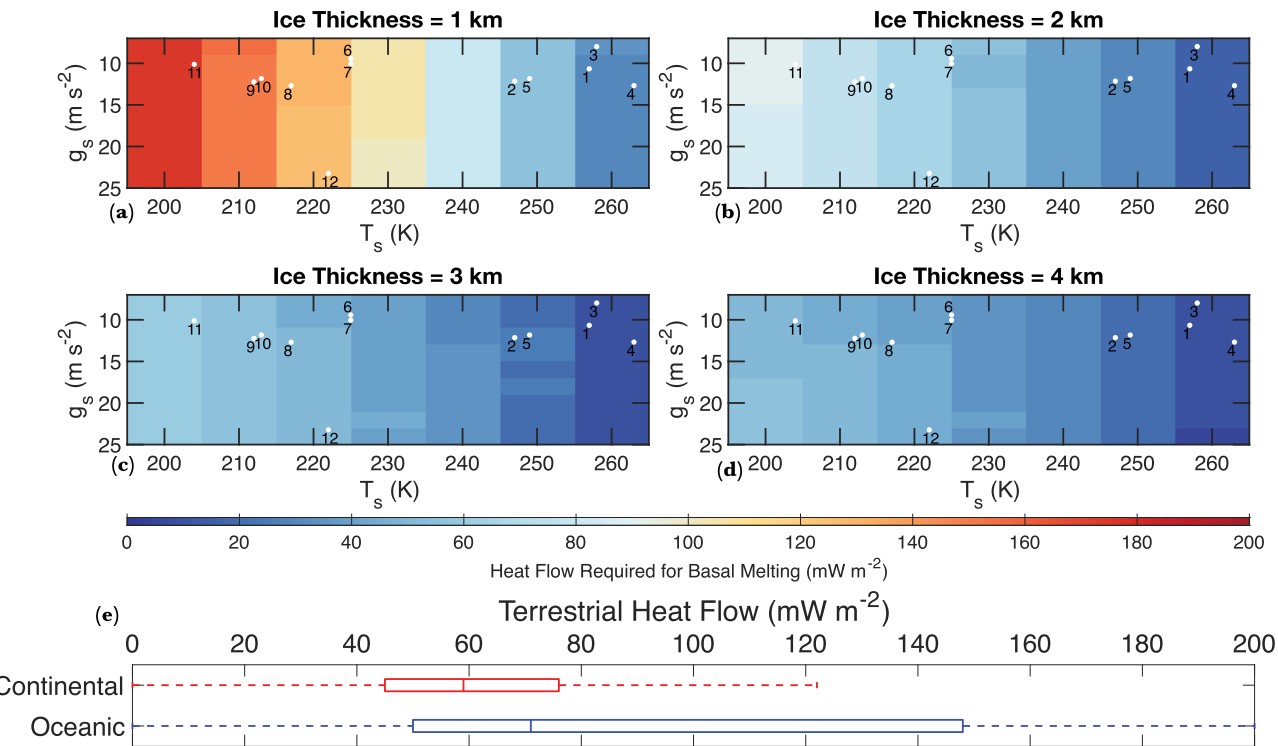

**Fig. 4 | Heat flow required for basal melting as a function of surface gravity and temperature for ice sheets 1–4 km in thickness. a** Heat flow required for basal melting as a function of surface gravity and surface temperature for a 1-km thick ice sheet. The white dots (the numbers correspond to the index number of planets in Table 1) show the approximate surface gravity and surface temperature of the exo-Earths considered in the study. **b–d** Same as (**a**), but for 2–4 km thick ice sheets. **e** A box and whisker diagram showing the heat flow distribution in the continental and the oceanic regions of the Earth. The box's lower and upper extent corresponds to the 25th and 75th percentile of the heat flow values, and the center corresponds to the median heat flow values.

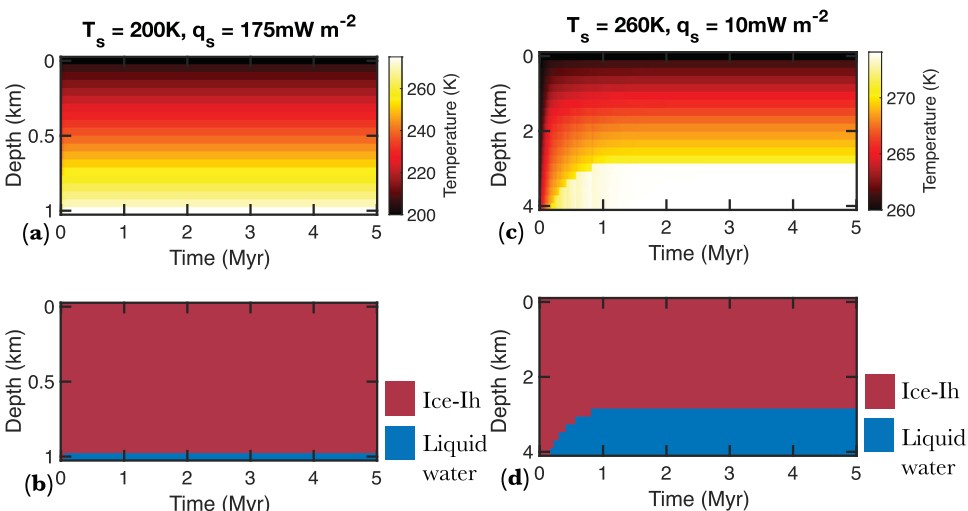

**Fig. 5 | Temperature and ice phases as a function of depth within a thick ice sheet on TRAPPIST 1 g and Kepler 442 b. a** Temperature distribution as a function of depth and time on TRAPPIST 1 g for a 1 km thick ice sheet, $T_s$ of 200 K, and heat flow of 175 mW m$^{-2}$. **b** Ice phase evolution as a function of depth and time on TRAPPIST 1 g over a 5-million-year time frame. **c** Temperature distribution as a function of depth and time on Kepler 442 b for a 4 km thick ice sheet, $T_s$ of 260 K, and heat flow of 10 mW m$^{-2}$. **d** Ice phase evolution as a function of depth and time on Kepler 442 b over a 5-million-year time frame.

extreme heat loss scenario in which the heat flow on Proxima Centauri B exponentially declines from the current Earth-like heat flow of 60 mW m$^{-2}$ to the current Mars-like heat flow of 30 mW m$^{-2}$ within a billion years. In this rapid heat loss scenario basal melt would only be stable for approximately 750 million years. In the second, moderate heat loss scenario, the same amount of heat flow declines over a

4-billion-year period. As shown in Fig. 8, basal melt in an exo-Earth with moderate heat loss can persist for a geologically prolonged period (>3 Gyrs). Supplementary Figs. 5 and 6 show similar results for the coldest exo-Earth in our study, TRAPPIST-1 g. The efficiency of heat loss on exo-Earths and its impact on basal melting is further assessed in the Discussion section.

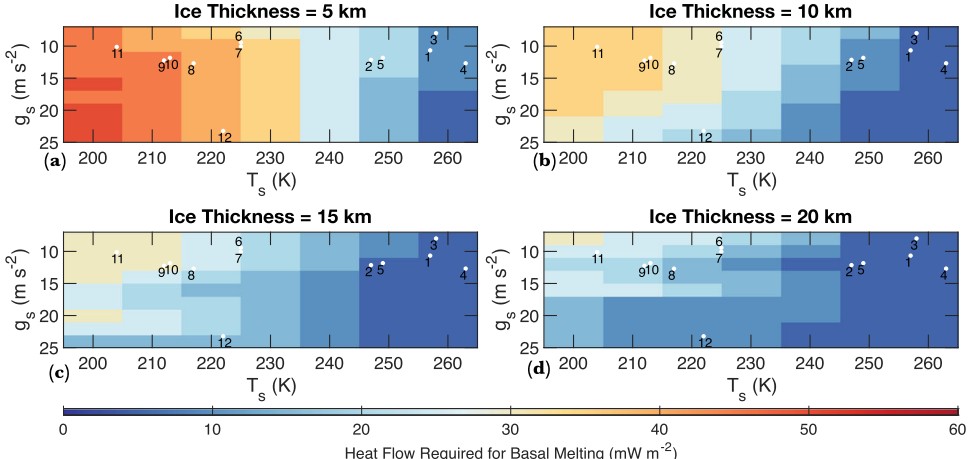

**Fig. 6 | Heat flow required for basal melting as a function of surface gravity and temperature for ice sheets 5–20 km in thickness. a** Heat flow required for basal melting as a function of surface gravity and surface temperature for a 5-km thick ice sheet. The white dots (the numbers correspond to the index number of planets in Table 1) show the approximate surface gravity and surface temperature of the exo-Earths considered in the study. **b–d** Same as (**a**), but for 10–20 km thick ice sheets.

## Discussion

The primary goal of this paper is to demonstrate the relative ease by which basal melting may be attainable on M-dwarf orbiting exo-Earths. While there are notable uncertainties about the presence and the volume of hydrospheres on these bodies, if even a handful of potentially habitable exo-Earths discovered so far (or in the future) were to contain thick (>few km) hydrospheres, then liquid water via basal melting may be present on those bodies with relatively modest heat flow. Our heat flow constraint required for basal melting, under the assumption that the estimated $T_{eq}$ is equal to the $T_s$, is an overestimate. The presence of greenhouse gases on any of these exo-Earths will raise the $T_s$, and therefore the heat flow required for basal melting would be notably lower than the estimates presented here.

While reasonable constraints on $T_s$ of exo-Earths can be placed based on the estimated $T_{eq}$, heat flow on exo-Earths is entirely unconstrained. The main source of long-term heat in a planet after the initial stage of accretion and differentiation is the radiogenic decay of isotopes of heat-producing elements with billion-year half-lives such as $^{40}K$, $^{232}Th$, $^{235}U$, and $^{238}U$ (i.e., geothermal heat). For example, approximately 80% of the Earth's present-day surface heat flow can be attributed to the decay of radioactive isotopes[53]. Radiogenic heat production as a function of age for cosmochemically Earth-like exoplanets suggests that exo-Earths similar in age to Earth should have a comparable heat production rate (H)[54]. However, there is a considerable degree of variations and uncertainties associated with the age of the M-dwarf systems considered in this study (Table 1). A comparison of the H values required for basal melting on the various exo-Earths to the age-dependent heat production values of cosmochemically Earth-like exoplanets[54,] is shown in Fig. 9. Despite the old age of some of the M-dwarf systems, the heat production rates on these exo-Earths may be sufficient for basal melting if they are cosmochemically Earth-like. The notable exceptions are TRAPPIST-1 f and TRAPPIST-1 g, where basal melting of thin ice sheets by geothermal heat alone may not be feasible given their old age (hence lower radiogenic heat production) and their relatively low surface temperature (also see Supplementary Fig. 2).

A comparison of the heat flow required for basal melting on various exo-Earths to the heat flow on moons and planets of our solar system may also allow us to further contextualize whether basal melting would be feasible on exo-Earths. The mean oceanic and continental heat flow on Earth is 101 and 65 mW m$^{-2}$, respectively (Fig. 4e); however, in active hot spots like Yellowstone and mid-oceanic ridges, the surface heat flow can exceed

1000 mW m$^{-2}$ [55]. On the Moon, heat-flow was measured to be $21 \pm 3$ and $15 \pm 2$ mW m$^{-2}$ at Apollo 15 and 17 landing sites respectively[56]. No direct measurements of the Martian surface heat flow are currently available; however, indirect remote sensing and models indicate heat flow up to 25 mW m$^{-2}$ [57,]. A first-order comparison of the heat flow required for basal melting on various exo-Earths to the heat flow of Earth provides further corroboration that basal melting may be entirely feasible on most exo-Earths with Earth like WMF (Fig. 9). For thick (>3 km) ice sheets, the heat production rate on exo-Earths can be ~50% that of Earth's, and basal melting can still occur (Supplementary Fig. 2). The heat production rate on Kepler 442 b can be ~5% that of Earth's, and basal melting could still occur. The thin horizontal lines in Fig. 9e show the range of heat flow for the Moon and Mars. In some planets like Kepler 442 b, Proxima Centauri B, and TRAPPIST-1e, even ice sheets that are 1-2 km thick may undergo basal melting with Mars-and-Moon like heat flow (Supplementary Fig. 2). Heat flows in excess of Earth's continental and oceanic average are only required to melt thin ice sheets on planets with extremely low $T_s$, such as TRAPPIST-1 g, GJ 667 C e, and Kepler-186 f (Fig. 4; Fig. 6; Supplementary Fig. 2). It is conceivable that the exo-Earths considered in this study are not cosmochemically Earth like and the geothermal heat flow on those exo-Earths may not be sufficient for basal melting. In such a scenario, tidal heating may provide an additional source of the heat on some exo-Earths around the habitable zone of M-dwarf stars[58]. For example, the age of the TRAPPIST-1 system is estimated to be $7.6 \pm 2.2$ Gyr;[59] thus, if geothermal heating has waned more than predicted by the age-dependent heat production rate assumed here[54], tidal heating could be an additional source of heat for basal melting on the TRAPPIST-1 system. On planets e and f of the TRAPPIST-1 system, tidal heating is estimated to contribute heat flow between 160 and 180 mW m$^{-2}$ [60,]. Thus, even if geothermal heating were to be negligible on these bodies, basal melting could still occur via tidal heating alone (Supplementary Fig. 2). However, for TRAPPIST-1 g, the mean tidal heat flow estimate from N-body simulation is less than 90 mW m$^{-2}$ [43,]. Thus, ice sheets thinner than a few kilometers are unlikely to undergo basal melting on TRAPPIST-1 g (Supplementary Fig. 2).

Due to the billion-year half-lives of the heat-producing elements responsible for planetary geothermal heat, meltwater created by basal melting may be sustained on exo-Earths for a prolonged period (Fig. 8; Supplementary Fig. 5). However, the longevity of basal melt relies on the cooling rate of the planet,

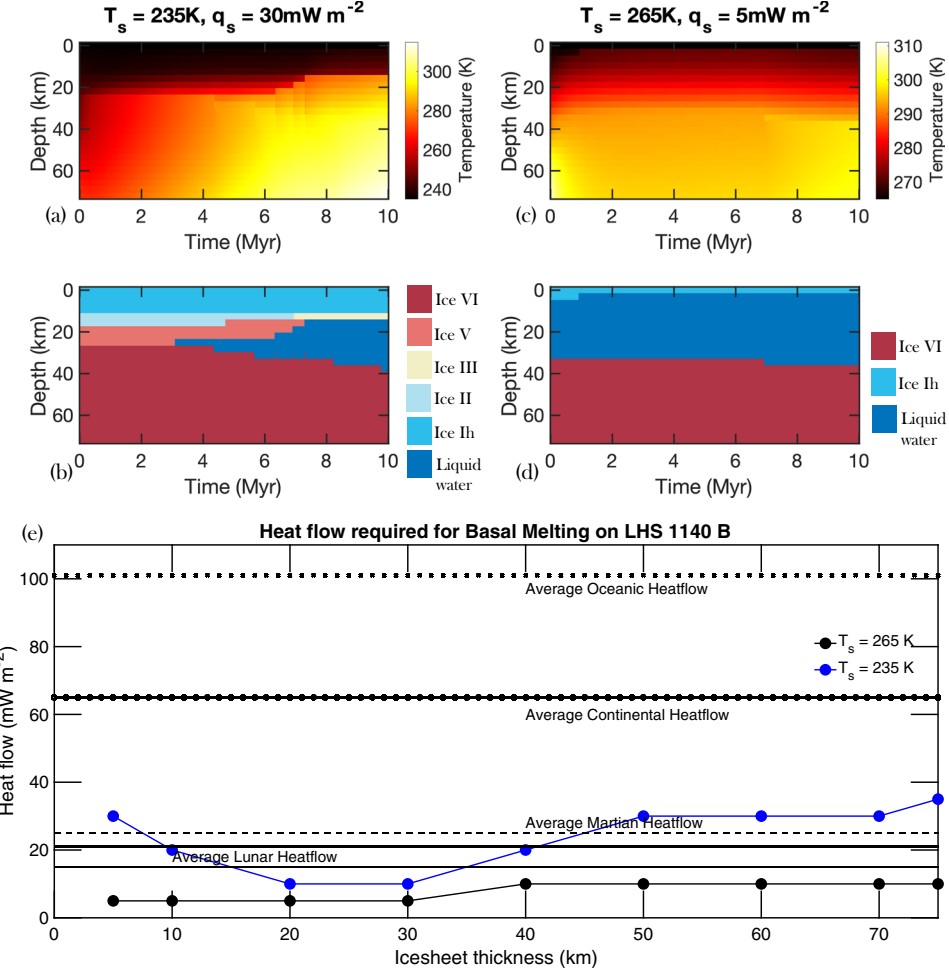

**Fig. 7 | Heat flow required for basal melting on super-Earths as a function of the ice sheet thickness. a** Temperature distribution as a function of depth and time on LHS 1140 b for a 70 km thick ice sheet, $T_s$ of 235 K, and heat flow of 30 mW m$^{-2}$. **b** Ice phase evolution as a function of depth and time on LHS 1140 b over a 10-million-year time period. The melt water underneath the high-pressure ice phases quickly migrates to shallower depths; however, the migration of buoyant liquid water is not modeled here. **c** Temperature distribution as a function of depth and time on LHS 1140 b for a 70 km thick ice sheet, $T_s$ of 265 K, and heat flow of 5 mW m$^{-2}$. **d** Ice phase evolution as a function of depth and time on LHS 1140 b over a 10-million-year time frame. **e** The black and blue lines show the average geothermal heat flow (y-axis) required for an ice sheet of certain thickness (x-axis) to undergo basal melting on LHS 1140 b assuming a surface temperature of 265 and 235 K. The black horizontal lines show the average heat flow observed on planets in the solar system. The presence of pressurized ice phase Ih with reduced melting point at 10–40 km depth enables basal melting with relatively low geothermal heat. Higher geothermal heating is required to melt thicker ice sheets as the melting temperature increases with ice thickness (see Fig. 2).

which is controlled by the mechanism and efficiency of heat transport from the interior of a planet to the surface[53]. Since the cooling histories of exo-Earths are unknown, here, we assess the longevity of basal melt by adopting an ad hoc cooling model in which the heat flow declines exponentially over a specified period. While the actual cooling history of the exo-Earths may be considerably different than the model used here, the goal here is to only investigate the first-order effect of planetary cooling on the feasibility of basal melting. Even when we consider an accelerated rate of planetary cooling, basal melt may be stable for a prolonged period (Fig. 8; Supplementary Fig. 5; Supplementary Fig. 6).

The density of liquid water is higher than water-ice Ih; thus, as long as the basal pressure does not exceed ~200 MPa, liquid water will remain stable at the base of the ice sheets. The interaction of planetary hydrospheres with silicate bedrock will also inevitably result in the incorporation of other soluble minerals/chemicals that have the potential for significantly lowering the freezing point of pure water and nutrients essential for sustaining habitable conditions[61,62]. However, quantifying the role of solutes such as NaCl or NH$_3$ on the feasibility of basal melting is currently not possible due to the lack of experimental

and theoretical data on these systems' thermodynamics at higher pressures[61,63]. Nevertheless, limited available data suggest that these solutes will have antifreeze effects on ice polymorphs of similar magnitude at pressures up to 2 GPa[64,65]. Additionally, the inclusion of solutes lowers the heat capacity of solutions with increasing concentrations;[61] thus, we can expect increased liquid stability for the same heat flux when solutes are present. The results presented here are based on pure water thermodynamics and thus represent a conservative scenario that allows liquid water to form and be stable in the hydrospheres of ice-rich exoplanets.

The subsurface world of these exo-Earths might resemble the subsurface conditions found on Europa. In this class III habitat[66], water in subsurface oceans interacts with silicates. The ensuing water-rock interactions at the crustal interface may provide a variety of chemicals and energy that could play a role in the origin and sustenance of putative life forms at the ocean floor, akin to those found at hydrothermal vents on Earth[67]. Despite the high pressures present at the base of the ice sheets on super-Earths, it may not be a limiting habitability agent as life on Earth has been observed at subduction forearc with pressure exceeding 340 MPa[68], and piezotolerant strains of

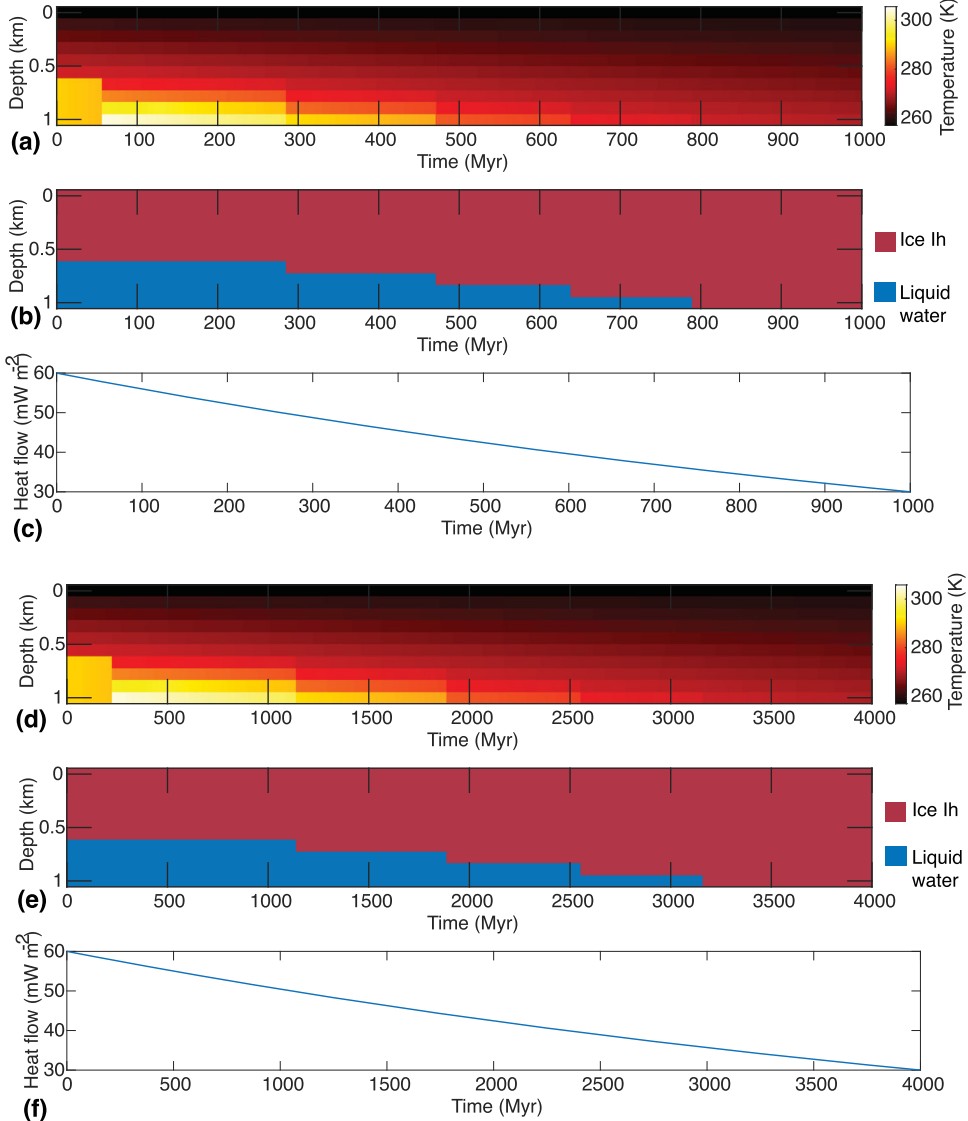

**Fig. 8 | Temperature and ice phases as a function of depth within a 1-km thick ice sheet on Proxima Centauri B, assuming exponentially declining heat flow.** **a** Temperature distribution as a function of depth and time on Proxima Centauri B for a 1 km thick ice sheet assuming $T_s$ of 257 K and variable heat flow over a billion years. **b** Ice phase evolution as a function of depth and time on Proxima Centauri B over a billion years. **c** An ad hoc exponential function that models heat loss on Proxima Centauri B from 60 to 30 mW m$^{-2}$ over a billion years. **d** Temperature distribution as a function of depth and time on Proxima Centauri B for a 1 km thick ice sheet assuming $T_s$ of 257 K and variable heat flow over 4 billion years. **e** Ice phase evolution as a function of depth and time on Proxima Centauri B over 4 billion years. **f** An ad-hoc exponential function that simulates heat loss on Proxima Centauri B from 60 to 30 mW m$^{-2}$ over 4 billion years.

bacteria on Earth can survive brief exposures up to 2000 MPa[69]. The efficacy of geothermal heat sustained basal meltwater providing a habitable niche[70], is exemplified by the continual habitability on Earth during the Neoproterozoic era (600–800 Ma) which was marked by widespread glaciation[20].

An isolated ocean between layers of low density, low pressure ice and high density, high pressure ice may also form (Fig. 7), without chemical exchange (i.e. nutrient flux) with the rocky core, which may not be adequate for habitability[62]. However, liquid aqueous solutions generated by basal melting at the rock/high-pressure ice mantle boundary will be buoyant and will permit the transportation of important life sustaining solutes/nutrients from the rocky core to the interglacial ocean. These bodies would classify as class IV habitat with liquid water layers between two ice layers[66], such as the internal models of Ganymede and Callisto. In some scenarios, the melt is only observed within the ice layer and not at the base of the ice sheet (Fig. 7). Still, the absorptive properties of the frozen surface ice layer

would shield basal melt environments from X-ray, extreme, and far UV radiation. Thus, basal melting provides a potentially habitable environment for cool, terrestrial planets orbiting M-dwarfs. It may also provide an environment isolated from the active and variable radioactive properties of M-dwarfs, which have long been a concern for the habitability of these planets.

## Methods

### Thermal evolution of ice sheets

Underneath thick planetary glaciers, the melting point of ice varies due to both pressure and ice phase. Due to the estimated high surface gravity on LHS 1140 b, Proxima Centauri B, and some of the planets in the Trappist system, water-ice can experience extreme pressures and temperatures at depth and evolve into high-pressure ice phases[61]. We use the code Sea-Freeze (https://github.com/Bjournaux/SeaFreeze)[71] in conjunction with a heat transport model to investigate the temporal and depth-dependent evolution of the thermodynamic and elastic

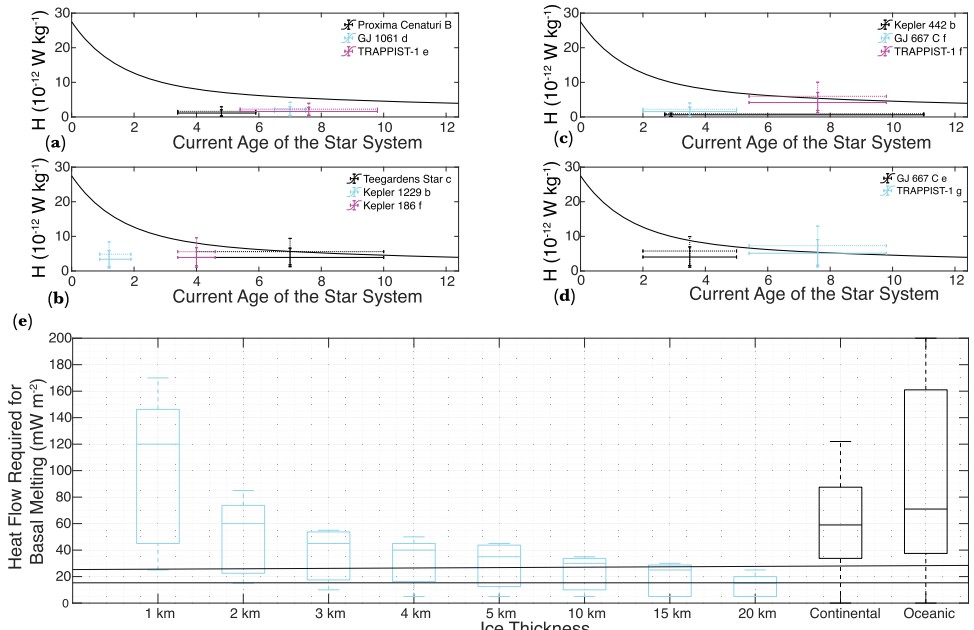

**Fig. 9 | Radiogenic heat-production rate required for basal melting compared with the age-dependent heat production rate of cosmochemically Earth-like planets. a** The black curve shows the radiogenic heat production rate as a function of time for cosmochemically Earth-like planets. The vertical extent of the error bars shows the heat production rate required for basal melting of ice sheets 1–20 km in thickness (also see Supplementary Fig. 2) on Proxima Centauri B, GJ 1061 (**d**), and TRAPPIST-1 (**e**). The dotted error bars show the range of heat production required for basal melting assuming Earth-like core mass fraction. The horizontal error bar shows the uncertainty in the age of these star systems. **b**–**d** Same as (**a**) but showing the range in the heat production rate and ages of various other exo-Earths listed in the legend. **d** A boxplot graph showing the range of heat flow required for basal melting on various exo-Earths as a function of the ice-thickness compared to the heat flow on Earth's continents and oceans. The two thin, horizontal black lines show the range of heat flow on the Moon (lower line) and Mars (upper line). Source data are provided as a Source Data file.

properties of water and ice polymorphs in the 0–2300 MPa and 220–500 K range.

Ice is compressive such that a large increase in pressure with depth can lead to a significant adiabatic increase of temperature with depth. The initial condition adiabatic temperature gradient implemented in all our simulations is given by the following:

$$\left(\frac{dT}{dz}\right) = \frac{\alpha_v g T}{c_p} \qquad (1)$$

where $\alpha_v$, $g$, $T$, and $c_p$ are thermal expansivity, gravity, temperature, and specific heat at constant pressure, respectively. An example of an adiabatic temperature profile assuming a surface temperature of 235 K and associated temperature-dependent thermal parameters is shown in Fig. 3. Sea-Freeze is a thermodynamic model which predicts the pressure and temperature dependent phase evolution of ice Ih and high-pressure ice polymorphs as well as an array of their material parameters[71]. Sea-Freeze does not model variations in thermal conductivity; thus, we use average thermal conductivities ($K_{avg}$) for each ice phase as listed in Supplementary Table 1.

The duration of possible glaciation on these bodies is unknown, but ice sheets on the night side of tidally locked planets may exist in a near-equilibrium state with the atmosphere for an extended period[72]. On Earth, widespread glaciation during the Neoproterozoic era (snowball Earth) may have prevailed for over 200 million years[20]. For a given heat flow, it only takes a few million years for the ice sheet to achieve thermal equilibrium, thus for computational efficiency, we run the thermophysical evolution models for 5 million years. Given the relatively short timescale of our models, we assume a steady-state heat production rate and neglect the secular cooling of the planet. In some special cases, we also model the thermophysical evolution of icesheets for a billion years to ascertain the stability of basal melt. In these

scenarios, we assume that the heat flow declines exponentially over a specified period.

The goal is to solve for the physical (melting) and thermal (heat transport) evolution of a thick, multilayered ice sheet containing Ice Ih, high-pressure ices, and any potential melt layers. To do so we solve a modified version of the 1-D thermal diffusion equation that accounts for the potential existence of three distinct heat transport regimes within the ice sheet. These regimes focus on representing the initial production of melt from heat transfer over the specified time intervals, and do not account for additional factors such as salinity prevalence and the buoyancy effects associated with viscosity contrasts at the resulting phase boundaries. Regime I is characterized by classic thermal diffusion in a solid and describes the heat transport in solid regions of the ice sheet that are not undergoing solid state convection. This includes ice layers that are too thin to undergo solid state convection as well as the conductive boundary layers of ice layers that are undergoing solid state convection (see below for the calculation of these boundary layer thicknesses). Conservation of energy in these regions is described by

$$\rho c_p \frac{\partial T}{\partial t} = \frac{\partial}{\partial z}\left(k \frac{\partial T}{\partial z}\right) \qquad (2)$$

where $\rho$ is density, $c_p$ is specific heat, $T$ is temperature, $t$ is time, $z$ is the vertical spatial coordinate, and $k$ is thermal conductivity[53]. The pressure and temperature dependence of density and specific heat are iteratively calculated using Sea-Freeze. The ice-phase dependency of thermal conductivity is updated based on the values reported in Supplementary Table 1.

Regime II is characterized by solid ice layers that are undergoing solid state convection. If there exist significant temperature-dependent density differences between the top and bottom of an ice

layer heated from below the layer is susceptible to solid state convection[73,74]. To determine if any portion of an ice layer is undergoing solid state convection we check to see if the layer's Rayleigh number (Ra) exceeds a preset critical Rayleigh number (Ra$_c$ = 1000), a common approach in solid state mantle convection models[75,76]. The Rayleigh number can be described as

$$\text{Ra} = \frac{\alpha_\upsilon \rho^2 c_p g \triangle T h^3}{k\eta} \qquad (3)$$

where $\Delta T$ is the temperature difference between the top and bottom of the ice layer, $h$ is the thickness of the ice layer, and $\eta$ is the viscosity of the top of the ice layer[53]. We additionally allow for the possibility of stagnant lid convection by testing whether any regions within each ice layer can undergo solid state convection. We assume ice viscosity follows a temperature dependent Arrhenius law[77]

$$\eta = \eta_0 \exp\left( A\left( \frac{T_m}{T} - 1 \right) \right) \qquad (4)$$

where $\eta_0$ is a characteristic viscosity (values for each ice type listed in Supplementary Table 1), $A$ is a constant coefficient here taken to be 7.5 (in line with ref. 77) and $T_m$ is the melting temperature of the ice (calculated by SeaFreeze).

When a layer of ice is undergoing solid state convection it will retain transitional conductive boundary layers at both its upper and lower interface, the thicknesses of which can be described by[75,76,78]

$$\delta_{\text{upper}} = h\left( \frac{\text{Ra}_c}{\text{Ra}} \right)^{1/3} \qquad (5a)$$

$$\delta_{\text{lower}} = h\left( 0.28\text{Ra}^{-0.79} \right)^{1/3} \qquad (5b)$$

Ice within these boundary layer regions will not be convecting and heat transport in these layers will be governed by Eq. 2. Heat transport in ice undergoing solid state convection is described by an amplified diffusion equation

$$\rho c_p \frac{\partial T}{\partial t} = \frac{\partial}{\partial z}\left( \text{Nu} \times \text{k} \frac{\partial T}{\partial z} \right) \qquad (6)$$

where Nu is the Nusselt number for the top layer of the convecting region (Nu = (Ra/Ra$_c$)$^{1/3}$)[75,76].

Regime III describes the heat transport in melt (liquid water) layers. The amplified thermal diffusion approach of Eq. 6 is applicable in regions where the convective timescale is on the order of or less than the temporal discretization timescale ($O$(10,000 years)), and thus is appropriate in high viscosity ice layers undergoing solid state convection. However, in low viscosity liquid water layers where the convective time scale is much less than 10,000 years an alternate approach is needed. Given the significant length of our temporal discretization ($O$(10,000 years)), we assume that all liquid water layers will be well mixed throughout the duration of our simulations. To accomplish this, during each model time step we simulate thermal diffusion into and out of the liquid water layer governed by Eq. 2. That is, thermal diffusion is simulated throughout the entire domain governed by either Eq. 2 or Eq. 6, where the only regions described by Eq. 6 are regions of ice layers undergoing solid state convection that are not within their conductive boundary layers. Liquid water layers are simulated as purely conductive (Eq. 2). We implement an implicit Euler finite discretization scheme to solve for the thermal diffusion throughout the ice sheet, subject to a constant Dirichlet surface temperature boundary at the top of the ice sheet and a Neumann geothermal heat flux boundary condition at the base of the ice sheet[23,26].

After the solution has stabilized, we calculate the enthalpy of each discretized layer within the liquid water layers (typically enthalpies at the base of the layers are higher because heat has been conducted into the liquid layer base and out of the top of the layer)

$$H_j = c_j T_j \qquad (7)$$

where $H_j$ and $c_j$ are the enthalpy and specific heat of the $j$th discretized layer. The enthalpies of a liquid layer are summed and then evenly redistributed throughout the liquid layer to simulate mixing. Enthalpy was employed, as opposed to temperature, to ensure conservation of energy during mixing given depth dependent specific heats. The temperature profile within the melt layer can easily be recovered by solving Eq. 7 for $T$.

The result is an energy conserving vertical temperature profile within the ice governed by Regimes I-III; thermal diffusion in solid ice layers not undergoing solid state convection (thin ice layers and conductive boundary layers), Nu-Ra parameterized convection within ice layers undergoing solid state convection, and thorough mixing of liquid water layers. After each temporal iteration of the thermal model the resulting temperature profile is fed to the SeaFreeze code to determine the new phase stratigraphy of the ice sheet (i.e., if any phase transformations between ice types or melting has occurred), and the material properties of the ice sheet are updated prior to the next model time step accordingly. This process is then iterated for the duration of the five-million-year simulations.

### Heat production rate

The total heat flow from the interior of a planet ($Q$) can be estimated by multiplying the surface area of the planet ($A$) by the mean heat flow (q) required for basal melting. The mean heat generation per unit mass ($H$) is then given by

$$H = \frac{Q}{M} \qquad (8)$$

where $M$ is the mass of the planet. Due to the lithophilic nature of heat-producing elements, they are assumed to be absent in the metallic core of a planet; thus, ideally, $H$ should be calculated only for the silicate portion of a planet. However, since the size and mass of the core of most exo-Earths are unknown, we compute $H$ for exo-Earths, assuming Earth-like core mass fraction of 30%. The $H$ required for basal melting is compared with the age-dependent radiogenic heat production model for cosmochemically Earth-like planets[54].

### Reporting summary

Further information on research design is available in the Nature Portfolio Reporting Summary linked to this article.

## Data availability

The mass, radius, and the surface gravity estimate of the exo-Earths considered here can be found at (https://exoplanetarchive.ipac. caltech.edu/) and also in the Source Data files. The terrestrial heat flow data used to create Fig. 4e can be found at (http://heatflow.org/). The time-dependent radiogenic heat production rate shown in Fig. 9 is included as supplementary information of the following publication (https://doi.org/10.1016/j.icarus.2014.08.031) and can also be found in the Source Data files. The datasets generated during and/or analysed during the current study are available from the corresponding author on reasonable request. Source data are provided with this paper.

## Code availability

SeaFreeze was used to estimate the thermodynamic and elastic properties of water and ice and can be found at the following location (https://github.com/Bjournaux/SeaFreeze). The codes used to model

the temporal and depth-dependent evolution of the ice sheets can be found here (https://github.com/lujuojha/ExoPlanetsBasalMelting) and (https://zenodo.org/record/7331987#.Y3ud4uzMJJF).

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

## Acknowledgements

L.O. was supported by a startup grant from Rutgers University. BJ was supported by the Habitability of Hydrocarbon Worlds: Titan and Beyond nodes of NASA Astrobiology Institute (08-NAI5- 0021 and 17-NAI82-17) and the NASA Solar System Workings grant 80NSSC17K0775.

## Author contributions

L.O. conceived the project and performed the modeling with significant aid from B.T. and J.B. L.O., J.B., B.T., and B.J. designed the heat flow models and aided in the interpretation of the results. B.J. aided in the thermophysical evolution, SeaFreeze models, and interpretation of the results. G.M.D. aided in the interpretation of the results. L.O. wrote the paper with significant feedback from all other authors.

## Competing interests

The authors declare no competing interests.
