## [Peer Review File · Nature Communications]

REVIEWER COMMENTS

Reviewer #1 (Remarks to the Author):

Review of “Liquid Water on Cold Exo-Earths via Basal Melting of Ice Sheets” by L. Ohja and others.

Key results

This paper presents the results of a model-based study which shows that deep (kilometres-depth) ice sheets could lead to the presence of liquid water on so-called cold exo-Earths in orbit around M dwarf stars beneath, or within, the ice sheets. The key finding is that geothermal heat fluxes required vary with the planets’ estimated surface temperatures, mass, and water content, but are well within the range of heat fluxes exhibited by terrestrial planets in the solar system, suggesting such subglacial ‘oceans’ may be quite common on rocky planets in orbit around M class stars. The shielding effect of the overlying ice layer could protect any putative life from cosmic and/or solar radiation, further making such oceans possible locations for the evolution of life. Overall, I feel the paper is interesting, and could move to publication, though I also feel it does need some rather major revisions.

Validity

The methods use the estimated characteristics (gravity, surface temperature, water mass fraction and hence possible ice thickness) for a set of 12 cold exo-Earths as the input to a two-stage modelling process. Published, open-source code (“SeaFreeze”, Journaux et al. 2019) is first used to calculate the initial thermodynamic properties (phase boundaries, density, specific heat, bulk modulus, thermal expansivity) of the inferred ice layer, and then a new 1-dimensional model for the thermal evolution of the ice is used to calculate the long-term (5 million year) equilibrium temperature when a range of geothermal heat fluxes are applied. The model allows for different layers with different dominant heat flux mechanisms (diffusive versus layers exhibiting solid state convection), the formation of conductive boundary layers above and below such layers, and assumes any liquid layers are fully mixed (given the model time step ($\sim 10^4$ years) under investigation) and so transfer heat using via conduction. The temperature/depth calculations are then fed back to the seafreeze code to update the phase stratigraphy, and the combined model iterated for the duration of the simulations. One possible omission from this framework is the possible presence of various salts in the water, but given the other uncertainties in the planetary conditions, the implicit assumption of pure water is not unreasonable.

Significance

Overall, I feel the results are significant. Given the growing number of exoplanets now detected, the large number of M class stars, and the findings of the paper itself, showing that relatively small geothermal heat fluxes are needed for melt to occur, the results suggest that sub-ice oceans may be quite common in the Milky Way galaxy. This in turn could have implications for potential evolution of life in other solar systems. One caveat could be the lifespan of M class stars, which might mean that (notwithstanding the long half-lives of the main heat-producing radionuclides) some of the planets may be orbiting very old stars, and may have very limited residual geothermal heat. I am not an expert on the age distribution of M class stars, but this might be worth commenting on in the paper given the argument around their long lives, which could make some of the systems much older than the solar system.

Analytical approach

The overall approach is justified given the uncertainties in estimates of the planetary conditions. However, I feel that the overall approach to this uncertainty is limited. There is some very limited acknowledgement of the range in these estimates in the paper (around line 134), but the main results are presented for a 'snapshot' of each planet (or for Figure 2, with two different inferred ice thicknesses and heat fluxes). It would, I think, be instructive to consider a denser set of experiments within the overall parameter space which the 12 planets exist within, so (for instance), to perform more runs within a 4-d space for planetary mass/gravity, inferred ice thickness, surface temperature and geothermal heat flux. The paper does already do some of this – such as the information shown in Supplementary Figure 2 for varying ice thicknesses on the different planets, but I think this could be expanded to consider more controls, acting together and separately, and presented in 'parameter space' rather than planet-by-planet. This would allow the authors to start to show more clearly the relative influence of each variable on the heat flux needed, and therefore the overall likelihood of water given the distribution of planetary mass and temperatures observed.

Suggested improvements and Clarity and Context

Here are where the biggest issues with the paper lie. Ultimately, it does not conform to the required layout of a Nature Communications article. I assume it has been transferred by editorial policy from a different Nature portfolio journal without adjusting the format. Sign-posting the article with section headings as per the journal format requirements would be a big help, but I feel the issues are deeper rooted. I feel a key issue is the lack of introduction to the planets chosen themselves; a table of their properties is just provided in the SI, but I think it is central to the paper itself, and also would

fit into the idea of a broader set of experiments which I suggest above, with the various planets 'fitting' into that parameter space (a bit like Supplementary Figure 1, which I like, though I have a suggestion for that figure below). Given more information on the planets, the choice of a 5km ice sheet for the 'default' runs becomes more justifiable – otherwise it feels rather arbitrary, especially given the information in table S1 (where, incidentally, the GEL column needs a unit – I assume it's km?). The paper as it stands feels quite 'introduction-heavy', with paragraphs 1-4 being introductory, but with 3, 4 and even 6 containing methodological aspects. Paragraph 5 is the main results paragraph, but most of the detail is actually in the supplementary information and I'd say at least Supplementary figure 3 should be in the main text as the results from the phase calculations within the model. Paragraphs 5-9 then contain rather a mix of results and discussion, which for clarity and to follow the journal guidelines should be disentangled to make the distinction clearer.

I hope that by revising the overall structure of the paper, the use of subheadings for different sections and the overall longer format (5000 words) allowed for Nature Communications articles, the article can move forward to publication, as I do feel the overall findings are interesting and significant.

Given the substantial revisions I am suggesting, I am not going to go through line-by-line, but I do have a few more specific suggestions which I hope are helpful.

Line 66 'above freezing' is very vague. Even in the abstract, I think this should say something like 'above the pressure-controlled freezing point of water' or words to that effect.

Line 81-82 is very oddly worded – I assume the paper is arguing that insufficient greenhouse warming occurs on many of the planets discussed making subglacial water the main way to produce liquid water on such planets.

Line 99 – I think a better reference for the distribution of terrestrial subglacial lakes in general is Livingstone, S. J. et al. "Subglacial lakes and their changing role in a warming climate". *Nat. Rev. Earth Environ.* 3, 106–124 (2022).

Line 108 – I'd suggest referencing Lauro, S. E. et al. "Multiple subglacial water bodies below the south pole of Mars unveiled by new MARSIS data". *Nat. Astron.* 5, 63–70 (2021) in addition.

Figure 1. I think a key is needed for the ice phases (which I also feel need a little more discussion in the introduction). Giving the value for g used to calculate the pressure at depth in the caption would also help show the context.

Figure 3 – I don't think given the table of % values really adds much – it's not hard to place the figures in a terrestrial context. I'm also not sure why some of the text is white, and some black? Is that just related to the intensity of the blue colouring? This comment also applies to Supplementary Figure 2.

Table S1 and Supplementary Figure S1 – should move to the main text I think. For figure S1, I'd suggest the colourbar is moved to be horizontal below the figure, as it would seem to relate to planetary mass (the x-axis), not equilibrium temperature (the y-axis).

Reviewer #2 (Remarks to the Author):

Review of manuscript NCOMMS-22-24296-T: Liquid Water on Cold Exo-Earths via Basal Melting of Ice Sheets, by Ojha et al.

Reviewed by S. Clifford.

Summary comments:

The authors identify the minimum thermal and pressure conditions under which liquid water might persist beneath ice covers of varying thicknesses for 12 examples of exo-Earths. Given the broad scope of the paper, the discussion might be a better fit to a journal with less restrictive page limits. However, the findings are important and original and indicate that for planets with thick hydrospheres, there may exist a potentially complex stratigraphy of habitable liquid water environments at depth (e.g., authors' Fig. 1). While I believe the research would benefit from a longer and more thorough discussion, I think the manuscript as currently written warrants publication in Nature Communications following the minor revisions identified below.

Specific comments:

Ln 127: the acronym 'WMF' is used before it is defined in line 132 (water mass fraction).

Ln 207: The analysis focuses on the stability of pure liquid water, which is an understandable and conservative simplification. However, any hydrosphere that is in physical contact with a lithic sea floor will inevitably result in the incorporation of other soluble minerals/chemicals that have the potential for significantly lowering the freezing point of liquid water (which would allow liquid water to persist at shallower depths than those calculated by the authors). While the authors make reference to the role that such solutes may play in creating conditions for "the origin and sustenance of putative lifeforms", a clearer acknowledgement of their potential influence on freezing point depression should also be made.

Lns 257-259: "In some scenarios, the melt is only observed within the ice layer and not at the base of the ice sheet (Supplementary Figure 7)." In looking at Supplementary Figure 7, there aren't any indications of the extent of Ices II-V (which are identified in the key) as a function of depth, so it just appears as though a liquid water ocean exists between Ice Ih and VI. I think the relationship of liquid water to the extent of these other ices needs to be more clearly illustrated.

Lns 319-326: duration of possible glaciation. For tidally-locked planets, it's difficult to see how there would be any temporal variation in the extent of glaciation aside from that associated with the long-term decline in geothermal heat flow – so the relevance of references to Earth and Mars here is unclear. In the case of Mars, the age of the ice at depth, within the SPLD, is generally thought to be on the order of ~1-3.5 billion years, so citing the inferred surface age isn't really relevant.

Since the thermal and pressure regimes that I'm most familiar with relate to Ice Ih, I'm not qualified to comment on the methods and results associated with Ices II-VI.

Reviewer #3 (Remarks to the Author):

This paper gives valuable testing and overview of the possibility of habitable environments on Earth- to Super-earth-sized exoplanets outside the usual Habitable Zone of M-stars, which also provides shielding against their energetic radiation in the early evolutionary phases. The author derives the conditions for the existence of liquid water beneath and within thick glacial sheets. Being in contact with the crust such sub- or intra-glacial lakes may provide attractive prospects for the evolution of life, somewhat similar to Europa and Enceladus in the Solar System, i.e. geothermal heat produced by tidal forces.

One obvious question is the role of radioactive elements. The author states that “subglacial oceans may persist on exo-Earths for a prolonged period due to the billion-year half-lives of the heat-producing elements responsible for geothermal heat” [67-68] – how important is this mechanism in producing subglacial liquid water without tidal heating (e.g. in planets too far from their hosts)? And vice versa, can tidal heating maintain basal melting on planets that do not have enough geothermal radioactive heat production? ([159-162]).

Running comments:

62. before "devoid of adequate greenhouse warming" - add "outside the Habitable Zone and".

75. [1] (Korappapu 2013) - it would be nice to add a more recent ref, also relating to sunlike planets, eg. Bryson et al 2021 AJ 161 36.

76-77. lifetimes of hundreds of billions of years - this is a common yet meaningless argument, lifetime (like our sun) is quite enough and M dwarf has no advantage in this respect.

82- 84. "X-rays and UV ... more flaring...presents challenges for the surface habitability of these planets" - this too is not a very strong reservation, as for aquatic life 20 cm of water can effectively shield these radiations, and water may be retained or regained by M-dwarf planets in spite of the early violent evolutionary epoch of their host, eg. Gale and Wandel 2017 IJA 16 1.

94. “basal melting may provide an alternative means of forming liquid water in a subsurface environment shielded from high-energy radiation (Fig. 1)” - According to Fig (1), a thick intraglacial ocean is present between the Ih and V layers. It is also well protected from energetic radiation. It should at least be motivated, why this is not enough for habitability and what is the added contribution of a basal melting layer (e.g. could the main ocean exist without a basal melting layer?)

127. WMF should be defined here rather than in [132]

154. If the masses are similar to Earth (or 6 times larger in the case of 1140b) why are only values up to 60 mW/m² considered? In [148] it is stated that Earth's heat flow is 60-100 mW/m².

204. ex Earths

216. Fig.3 – Heat production rate is monotonously decreasing with thickness in the range of 1-10 km. How is this consistent with Fig. 1, which indicates a non-monotonous temperature dependence of the ice-liquid boundary in this range?

218. GEL is undefined in the main ms (only in the Supp. File [168])

237. Supp/ Fig 7 shows that actually there is hardly any evolution

239 “due to the pressurized ice at the base, which has a reduced melting point (Fig. 1).” – how is this shown in Fig 1? Same question for [249].

324. What happens after the 5-million-year limit? This is a short time compared with the evolution of life. The author needs to comment on this. However, Fig. 2 indicates a steady state is reached after less than 2 million years for Proxima Cen b. Is this typical also for the other planets and for different parameter values?

325. Assuming a steady state and neglecting secular cooling may underestimate the heat input required in order to keep liquid water for extended biological periods (billions of years).

337. Give ref or evaluate eq 2

339. c should be cP.

341. "...thermal conductivity are iteratively calculated using Sea-Freeze or Table 1" – how does this comply with the above statement [313] that Sea-Freeze does not model variations in thermal conductivity?

350. Give ref or evaluate eq. 3

380. Apparently eq. 6 gives values only slightly different from eq. 2 unless R_a is very different from R_{ac} . What is the expected range of R_a ? What is typically the difference between the model including solid-state convection and calculating using only eq. 2, for all regions?

395-396. Give ref. or define/explain.

402. Define c_j (c_{Pj} ?)

425. The metallic core of a planet cannot contain a significant fraction

426 of the heat-producing element – why? also, should be plural (elements).

KEY

BOLD: Reviewer's Query

Italicized: Response to the Reviewers

Italicized & Underlined: Revised Text from the New Version

Reviewer #1 (Remarks to the Author):

Review of “Liquid Water on Cold Exo-Earths via Basal Melting of Ice Sheets” by L. Ohja and others.

Key results

This paper presents the results of a model-based study which shows that deep (kilometres-depth) ice sheets could lead to the presence of liquid water on so-called cold exo-Earths in orbit around M dwarf stars beneath, or within, the ice sheets. The key finding is that geothermal heat fluxes required vary with the planets' estimated surface temperatures, mass, and water content, but are well within the range of heat fluxes exhibited by terrestrial planets in the solar system, suggesting such subglacial ‘oceans’ may be quite common on rocky planets in orbit around M class stars. The shielding effect of the overlying ice layer could protect any putative life from cosmic and/or solar radiation, further making such oceans possible locations for the evolution of life. Overall, I feel the paper is interesting, and could move to publication, though I also feel it does need some rather major revisions.

Dear Reviewer,

We thank you for your constructive, detailed, and prompt review. We have significantly revised the paper per your suggestion and a point-by-point response to your review is presented below.

Validity

The methods use the estimated characteristics (gravity, surface temperature, water mass fraction and hence possible ice thickness) for a set of 12 cold exo-Earths as the input to a two-stage modelling process. Published, open-source code (“SeaFreeze”, Journaux et al. 2019) is first used to calculate the initial thermodynamic properties (phase boundaries, density, specific heat, bulk modulus, thermal expansivity) of the inferred ice layer, and then a new 1-dimensional model for the thermal evolution of the ice is used to calculate the long-term (5 million year) equilibrium temperature when a range of geothermal heat fluxes are applied. The model allows for different layers with different dominant heat flux mechanisms (diffusive versus layers exhibiting solid state convection), the formation of conductive boundary layers above and below such layers, and assumes any liquid layers are fully mixed (given the model time step ($\sim 10^4$ years) under investigation) and so transfer heat using via conduction. The temperature/depth calculations are then fed back to the seafreeze code to update the phase stratigraphy, and the combined model iterated for the duration of the simulations. One possible omission from this framework is the possible presence of various salts in the water, but given the other uncertainties in the planetary conditions, the implicit assumption of pure water is not unreasonable.

Indeed, the presence of various salts can significantly depress the melting point of water. However, as you stated, given the uncertainties associated with other parameters, we kept the modeling effort focused only on pure water. Furthermore, the focus of the paper as stated in our concluding paragraph is to “demonstrate the relative ease by which basal melting may be attainable on these cold, potentially icy

exo-Earths.” Brines, if present, would be even more likely to undergo melting on these cold exo-Earths, so we did not think including a discussion of melting brines would be a notable addition. Another unknown is if the laboratory measured reduction in the melting point of water due to salts scales linearly with pressure. Reviewer#2 also had a similar comment so we have added the following paragraph in the discussion section:

“[14] The interaction of planetary hydrospheres with silicate bedrock will also inevitably result in the incorporation of other soluble minerals/chemicals that have the potential for significantly lowering the freezing point of pure water and nutrients essential for sustaining habitable conditions [59, 60]. However, quantifying the role of solutes such as NaCl or NH₃ on the feasibility of basal melting is currently not possible due to the lack of experimental and theoretical data on these systems’ thermodynamics at higher pressures [59, 61]. Nevertheless, limited available data suggest that these solutes will have antifreeze effects on ice polymorphs of similar magnitude at room pressures up to 2 GPa [62, 63]. Additionally, the inclusion of solutes lowers the heat capacity of solutions with increasing concentrations [59]; thus, we can expect increased liquid stability for the same heat flux when solutes are present. The results presented here are based on pure water thermodynamics and thus represent a conservative scenario that allows liquid water to form and be stable in the hydrospheres of ice-rich exoplanets.”

Significance

Overall, I feel the results are significant. Given the growing number of exoplanets now detected, the large number of M class stars, and the findings of the paper itself, showing that relatively small geothermal heat fluxes are needed for melt to occur, the results suggest that sub-ice oceans may be quite common in the Milky Way galaxy. This in turn could have implications for potential evolution of life in other solar systems. One caveat could be the lifespan of M class stars, which might mean that (notwithstanding the long half-lives of the main heat-producing radionuclides) some of the planets may be orbiting very old stars, and may have very limited residual geothermal heat. I am not an expert on the age distribution of M class stars, but this might be worth commenting on in the paper given the argument around their long lives, which could make some of the systems much older than the solar system.

This is a great point. Indeed, the age of the M class stars can indeed be much older than our sun. We list the estimated age of all M-dwarf systems in Table 1, and model the decline in heat production using GCE models and further discuss the implication of their age on the feasibility of basal melting:

“Radiogenic heat production as a function of age for cosmochemically Earth-like exoplanets suggests that exo-Earths similar in age to Earth should have a comparable heat production rate (H) [51]. However, there is a considerable degree of variations and uncertainties associated with the age of the M-dwarf systems considered in this study (Table. 1). A comparison of the H required for basal melting on the various exo-Earths to the age-dependent heat production values of cosmochemically Earth-like exoplanets (e.g., [51]) is shown in Figure 9. Despite the old age of some of the M-dwarf systems, the heat production rate on these exo-Earths may be sufficient for basal melting if they are cosmochemically Earth-like. The notable exceptions are TRAPPIST-1 f and TRAPPIST-1 g, where basal melting of thin ice sheets by geothermal heat alone may not be feasible given their old age (hence lower radiogenic heat production) and their relatively low surface temperature (also see Supplementary Figure 2).”

We have also added the following text to address this important point along with Reviewer #3's concern on the effect of tidal heating:

"It is conceivable that the exo-Earths considered in this study are not cosmochemically Earth like and the geothermal heat flow on those exo-Earths may not be sufficient for basal melting. In such a scenario, tidal heating may provide additional source of the heat on some exo-Earths around the habitable zone of M-dwarf stars [55]. For example, the age of the TRAPPIST-1 system is estimated to be 7.6 ± 2.2 Gyr [56]; thus, if geothermal heating has waned more than predicted by the age-dependent heat production rate assumed here [51], tidal heating could be an additional source of heat for basal melting on the TRAPPIST-1 system [57, 58]."

Analytical approach

The overall approach is justified given the uncertainties in estimates of the planetary conditions. However, I feel that the overall approach to this uncertainty is limited. There is some very limited acknowledgement of the range in these estimates in the paper (around line 134), but the main results are presented for a 'snapshot' of each planet (or for Figure 2, with two different inferred ice thicknesses and heat fluxes). It would, I think, be instructive to consider a denser set of experiments within the overall parameter space which the 12 planets exist within, so (for instance), to perform more runs within a 4-d space for planetary mass/gravity, inferred ice thickness, surface temperature and geothermal heat flux. The paper does already do some of this – such as the information shown in Supplementary Figure 2 for varying ice thicknesses on the different planets, but I think this could be expanded to consider more controls, acting together and separately, and presented in 'parameter space' rather than planet-by-planet. This would allow the authors to start to show more clearly the relative influence of each variable on the heat flux needed, and therefore the overall likelihood of water given the distribution of planetary mass and temperatures observed.

Our choice to present specific example cases for the exo-Earths in the older version of the manuscript was simply due to the time it takes for us to run each individual model. We improved the code efficiency and did conduct some 'dense set of experiments' to assess the feasibility of basal melting in the overall parameter space. We have added two new figures (Fig. 4 and 6) where we solve for heat flow that allows basal melting as a function of surface temperature and gravity for icesheets of various thickness. Results from this exercise and discussion are also expanded.

Suggested improvements and Clarity and Context

Here are where the biggest issues with the paper lie. Ultimately, it does not conform to the required layout of a Nature Communications article. I assume it has been transferred by editorial policy from a different Nature portfolio journal without adjusting the format. Sign-posting the article with section headings as per the journal format requirements would be a big help, but I feel the issues are deeper routed. I feel a key issue is the lack of introduction to the planets chosen themselves; a table of their properties is just provided in the SI, but I think it is central to the paper itself, and also would fit into the idea of a broader set of experiments which I suggest above, with the various planets 'fitting' into that parameter space (a bit like Supplementary Figure 1, which I like, though I have a suggestion for that figure below). Given more information on the planets, the choice of a 5km ice sheet for the 'default' runs becomes more justifiable – otherwise it feels rather arbitrary, especially given the information in table S1 (where, incidentally, the GEL column needs a unit – I assume it's km?). The paper as it stands feels

quite ‘introduction-heavy’, with paragraphs 1-4 being introductory, but with 3, 4 and even 6 containing methodological aspects. Paragraph 5 is the main results paragraph, but most of the detail is actually in the supplementary information and I’d say at least Supplementary figure 3 should be in the main text as the results from the phase calculations within the model. Paragraphs 5-9 then contain rather a mix of results and discussion, which for clarity and to follow the journal guidelines should be disentangled to make the distinction clearer.

This paper was originally submitted to a different Nature family journal where formatting requirements were quite different. Then, the paper was directly transferred to Nature Communications. We did not have a chance to edit the stylistic content of the paper to conform to Nature Communications. In this version, we have revised the paper substantially (in terms of text and presentation) to conform to Nature Communications formatting requirement. We have added a section in the introduction to introduce the planets and our rationale for including them in this study:

[3] The feasibility of basal melting depends on the thermal and phase evolution of an ice sheet which is primarily governed by a planet’s surface temperature and geothermal heat. Other factors, such as surface gravity and hydrosphere thickness, can also play a major role in the feasibility of basal melting, given the pressure-dependent melting temperature of water ice (Fig. 2). Notably, if the pressure in the hydrosphere is in excess of 0.2 GPa, dense high-pressure ice polymorphs (II, III, V and VI) can form at the bottom trapping the interglacial ocean from access to vital solutes/nutrients limiting the interglacial ocean’s habitability potential. Basal melting, if present, would enable water-rock interaction and the formation of buoyant aqueous solutions that can rise through the high-pressure ice layer to feed the interglacial ocean enabling potential life-sustaining conditions (Fig. 2).

[4] Numerous M-dwarf orbiting exo-Earths with a high Earth Similarity Index (ESI) value, a multiparameter index that compares attributes such as mass, radius, and temperature of an exoplanet to Earth, have been discovered in the last decade (e.g., [5]) (Fig. 1; Table 1). These exo-Earths’ position within the CHZ of their parent stars also suggest that they could be water-rich planets. For example, the exo-Earth Proxima Centauri B orbits the star nearest to Earth. While the planet’s water content is unknown, a wide range of internal structure models for Proxima Centauri B suggests that it could be a water-rich planet (up to 50% by mass) [34]. However, the stability of liquid water on the surface of Proxima Centauri B is hard to reconcile with the relatively low T_{eq} of 257 K [35]. Del Ginio et al. (2019) used an atmospheric general circulation model (GCM) coupled with a dynamic ocean GCM to show that even with an atmosphere consisting of 10000 parts per million volume (ppmv) CO_2 and 2000 ppmv CH_4 , only a portion of the planet may be able to sustain liquid water. Thus, if Proxima Centauri B is a water-rich planet, as its internal structure model suggests [34], a significant fraction of the water may exist in the form of solid ice, where basal melting may be feasible.

[5] The internal structure models of several other exo-Earths suggest that they may be water-rich planets. For example, the internal structure model of the TRAPPIST-1 planetary systems suggest that they could potentially be more water-rich than Earth [36-38], although with notable uncertainty. Despite the strong far-UV photolysis of H_2O and large H_2 escape rates expected from M-dwarf orbiting exo-Earths, models suggest that TRAPPIST-1 planets may have retained a significant amount of water [39]. While liquid water may be stable on the surface of TRAPPIST-1 e with atmospheric H_2O alone, the other two planets (TRAPPIST-1 f, g) require greenhouse gases such as CO_2 and a thick atmosphere to sustain surface liquid water [40]. LHS 1140 B is a super Earth with an estimated mass 6 times than of Earth’s. A recent work utilized the internal structure model

developed by [41] to indicate that the water mass fraction (WMF) on LHS 1140 b could be as much as 4 % (80 times more than Earth), leading to an average global ocean of 779+/- 650 km. That study's 1% confidence level corresponds to a WMF = 0.007, still about 1.5 times more water than Earth [42]. The transmission spectrum of LHS 1140 b from the Hubble Space Telescope also shows tentative evidence of water, but future observations are needed to confirm this putative detection [43]. The exo-Earth GJ 1061 d receives a similar amount of energy as Earth receives from the Sun and may also be a water-rich planet [44]. GJ 667 C e and GJ 667 C f lie also within the CHZ of its host star and may be water-rich; however, they may still require a thicker atmosphere than Earth and greenhouse gases like CO₂ and CH₄ for the liquid water to be stable on their surface [45]. Kepler-442 b is a potentially water-rich exo-Earth with a high ESI value and potential for habitability [46]. Both Kepler 1229 b and 186 f have extremely low T_{eq} and thus would require a thick atmosphere with greenhouse gases to sustain liquid water [47]. In the absence of a thick atmosphere, a significant fraction of water on these two exo-Earths may exist in the form of ice. While there are significant uncertainties regarding the possible presence and the volume of the hydrosphere on these planets, even with an Earth-like water mass fraction (WMF) of 0.05%, these bodies could contain kilometers-thick global ice sheets (Table. 1), where basal melting may not only be possible but also play a significant role in habitability.

[6] In this work, we model the thermophysical evolution of ice sheets of various thicknesses and demonstrate that basal melting is likely prevalent on M-dwarf orbiting exo-Earths, even with modest, Moon-like geothermal heat flow. We show that thick subglacial oceans of liquid water can form and persist at the base of and within the ice sheets on exo-Earths for a prolonged period. Our findings suggest that exo-Earths resembling the snowball Earth or the icy moons of Jupiter and Saturn may be prevalent in the Milky-way galaxy.”

We have also (largely) rewritten the results, including sign-posting where appropriate and discussion sections to separate our interpretation from the model results.

I hope that by revising the overall structure of the paper, the use of subheadings for different sections and the overall longer format (5000 words) allowed for Nature Communications articles, the article can move forward to publication, as I do feel the overall findings are interesting and significant. Given the substantial revisions I am suggesting, I am not going to go through line-by-line, but I do have a few more specific suggestions which I hope are helpful.

Line 66 ‘above freezing’ is very vague. Even in the abstract, I think this should say something like ‘above the pressure-controlled freezing point of water’ or words to that effect.

Thank you for this suggestion. We have replaced ‘freezing’ in the abstract and in two other relevant sections to ‘above the pressure-controlled freezing point of water.’

Line 81-82 is very oddly worded – I assume the paper is arguing that insufficient greenhouse warming occurs on many of the planets discussed making subglacial water the main way to produce liquid water on such planets.

The intention here is to illustrate that given the low T_{eq} of these bodies, sufficient greenhouse warming would be necessary for liquid water to exist on the surface. However, even if these planets had sufficient greenhouse warming, surface habitability might still be an issue given the high surface radiation level these planets experience. We have rewritten this section:

“Even if the harsh effects of the M-dwarf stellar environment were absent, a significant fraction of the M-dwarf orbiting exo-Earths would still require substantial greenhouse warming for liquid water to be stable on the surface, given their relatively low equilibrium temperature (T_{eq}) (Fig. 1; Table 1). However, the efficacy of greenhouse warming depends on various factors such as albedo, cloud cover, greenhouse gas species, and their residence time in the atmosphere; parameters that are not well constrained for most Earth-sized exoplanets (e.g., [10]).”

Line 99 - I think a better reference for the distribution of terrestrial subglacial lakes in general is Livingstone, S. J. et al. “Subglacial lakes and their changing role in a warming climate”. Nat. Rev. Earth Environ. 3, 106–124 (2022).

Thanks. We have replaced the various references with the Livingstone et al. (2022).

Line 108 - I’d suggest referencing Lauro, S. E. et al. “Multiple subglacial water bodies below the south pole of Mars unveiled by new MARSIS data”. Nat. Astron. 5, 63–70 (2021) in addition.

Thanks. We have added Lauro et al. (2021).

Figure 1. I think a key is needed for the ice phases (which I also feel need a little more discussion in the introduction). Giving the value for g used to calculate the pressure at depth in the caption would also help show the context.

We do not completely understand this comment. Figure 1 (now Figure 2 in this version) does show key for the various ice phases, so we are unclear what the reviewer means by ‘key is needed’. We now provide the value for g used to calculate the pressure and depth in the caption and have added more discussion in the introduction.

Figure 3 - I don’t think given the table of % values really adds much - it’s not hard to place the figures in a terrestrial context. I’m also not sure why some of the text is white, and some black? Is that just related to the intensity of the blue colouring? This comment also applies to Supplementary Figure 2.

This was one of the figures informal reviewers (pre-submission) asked us to include to help contextualize the results. We think that the table % value helps contextualize what the heat flow values required for basal melting on exo-Earths means in comparison to Earth and other bodies of the solar system.

The change in text-color seems to be a default option when using MATLAB’s heatmap command. The author could not really find a way to change it, so we have added a line in the caption stating that the change in color is only for clarity.

Table S1 and Supplementary Figure S1 - should move to the main text I think. For figure S1, I’d suggest the colourbar is moved to be horizontal below the figure, as it would seem to relate to planetary mass (the x-axis), not equilibrium temperature (the y-axis).

We have moved the colorbar to be horizontal below the x-axis and moved both Table S1 and Supplementary Figure S1 to the main text.

Reviewer #2 (Remarks to the Author):

Review of manuscript NCOMMS-22-24296-T: Liquid Water on Cold Exo-Earths via Basal Melting of Ice Sheets, by Ojha et al.

Reviewed by S. Clifford.

Summary comments:

The authors identify the minimum thermal and pressure conditions under which liquid water might persist beneath ice covers of varying thicknesses for 12 examples of exo-Earths. Given the broad scope of the paper, the discussion might be a better fit to a journal with less restrictive page limits. However, the findings are important and original and indicate that for planets with thick hydrospheres, there may exist a potentially complex stratigraphy of habitable liquid water environments at depth (e.g., authors' Fig. 1). While I believe the research would benefit from a longer and more thorough discussion, I think the manuscript as currently written warrants publication in Nature Communications following the minor revisions identified below.

We thank Dr. Clifford for their constructive, detailed, and prompt review. A more detailed paper on this topic is currently in preparation, but we have also expanded the discussion of this paper significantly as per the request of Reviewer#1.

Specific comments:

Ln 127: the acronym 'WMF' is used before it is defined in line 132 (water mass fraction).

Thanks. Fixed now.

Ln 207: The analysis focuses on the stability of pure liquid water, which is an understandable and conservative simplification. However, any hydrosphere that is in physical contact with a lithic sea floor will inevitably result in the incorporation of other soluble minerals/chemicals that have the potential for significantly lowering the freezing point of liquid water (which would allow liquid water to persist at shallower depths than those calculated by the authors). While the authors make reference to the role that such solutes may play in creating conditions for "the origin and sustenance of putative lifeforms", a clearer acknowledgement of their potential influence on freezing point depression should also be made.

Indeed, the presence of various salts can significantly depress the melting point of water. However, as you stated, given the uncertainties associated with other parameters, we kept the modeling effort focused only on pure water. Furthermore, the focus of the paper as stated in our concluding paragraph is to "demonstrate the relative ease by which basal melting may be attainable on these cold, potentially icy exo-Earths." Brines, if present, would be even more likely to undergo melting on these cold exo-Earths, so we did not think including a discussion of melting brines would be a notable addition. Reviewer#1 also had a similar comment so we have added a new section on the effects of salts in the discussion section of the paper.

"The density of liquid water is higher than water-ice Ih; thus, as long as the basal pressure does not exceed ~200 MPa, liquid water will remain stable at the base of the ice sheets. The interaction of planetary hydrospheres with silicate bedrock will also inevitably result in the incorporation of other soluble minerals/chemicals that have the potential for significantly lowering the freezing point of pure water and nutrients essential for sustaining habitable conditions [59, 60]. However, quantifying the

role of solutes such as NaCl or NH₃ on the feasibility of basal melting is currently not possible due to the lack of experimental and theoretical data on these systems' thermodynamics at higher pressures [59, 61]. Nevertheless, limited available data suggest that these solutes will have antifreeze effects on ice polymorphs of similar magnitude at room pressures up to 2 GPa [62, 63]. Additionally, the inclusion of solutes lowers the heat capacity of solutions with increasing concentrations [59]; thus, we can expect increased liquid stability for the same heat flux when solutes are present. The results presented here are based on pure water thermodynamics and thus represent a conservative scenario that allows liquid water to form and be stable in the hydrospheres of ice-rich exoplanets.

Lns 257-259: “In some scenarios, the melt is only observed within the ice layer and not at the base of the ice sheet (Supplementary Figure 7).” In looking at Supplementary Figure 7, there aren't any indications of the extent of Ices II-V (which are identified in the key) as a function of depth, so it just appears as though a liquid water ocean exists between Ice Ih and VI. I think the relationship of liquid water to the extent of these other ices needs to be more clearly illustrated.

The presence of Ices II-V also depends on the temperature profile of the hydrosphere. This is illustrated in Figure 2 (b), where the presence of these ice phases is both temperature and pressure dependent. In the old Supplementary Figure 7, we start with a surface temperature of 265 K, so the temperature range required for the presence of Ices II-V is never reached in the deeper hydrosphere. In Supplementary Figure 3 of this manuscript, which assumes a surface temperature of 235 K, the presence and evolution of various other ice phases is shown. We have also added discussion on the paper about the fate of basal meltwater when higher pressure ices are present.

“Notably, if the pressure in the hydrosphere is in excess of 0.2 GPa, dense high-pressure ice polymorphs (II, III, V and VI) can form at the bottom trapping the interglacial ocean from access to vital solutes/nutrients limiting the interglacial ocean's habitability potential. Basal melting, if present, would enable water-rock interaction and the formation of buoyant aqueous solutions that can rise through the high-pressure ice layer to feed the interglacial ocean enabling potential life-sustaining conditions (Fig. 2).”

Lns 319-326: duration of possible glaciation. For tidally-locked planets, it's difficult to see how there would be any temporal variation in the extent of glaciation aside from that associated with the long-term decline in geothermal heat flow – so the relevance of references to Earth and Mars here is unclear. In the case of Mars, the age of the ice at depth, within the SPLD, is generally thought to be on the order of ~ 1-3.5 billion years, so citing the inferred surface age isn't really relevant.

This is a very good point. We have removed the discussion about Mars and simply state the following:

“The duration of possible glaciation on these bodies is unknown, but ice sheets on the night side of tidally locked planets may exist in a near-equilibrium state with the atmosphere for an extended period [70]. On Earth, widespread glaciation during the Neoproterozoic era (snowball Earth) may have prevailed for over 200 million years [20]. For a given heat flow, it only takes a few million years for the ice sheet to achieve thermal equilibrium, thus for computational efficiency, we run the thermophysical evolution models for 5 million years. Given the relatively short timescale of our models, we assume a steady-state heat production rate and neglect the secular cooling of the planet. In some special cases, we also model the thermophysical evolution of icesheets for a billion years to ascertain the stability of basal melt. In these scenarios, we assume that the heat flow declines exponentially over a specified period.”

We also added some new figures which assumes long term decline in heat flow. As long as the heat flow is above some threshold required for basal melting, the meltwater remains stable.

Since the thermal and pressure regimes that I'm most familiar with relate to Ice Ih, I'm not qualified to comment on the methods and results associated with Ices II-VI.

We thank you again for your helpful comments.

Reviewer #3 (Remarks to the Author):

This paper gives valuable testing and overview of the possibility of habitable environments on Earth- to Superearth-sized exoplanets outside the usual Habitable Zone of M-stars, which also provides shielding

against their energetic radiation in the early evolutionary phases. The author derives the conditions for the existence of liquid water beneath and within thick glacial sheets. Being in contact with the crust such sub- or intra-glacial lakes may provide attractive prospects for the evolution of life, somewhat similar to Europa and Enceladus in the Solar System, i.e. geothermal heat produced by tidal forces. One obvious question is the role of radioactive elements. The author states that “subglacial oceans may persist on exo-Earths for a prolonged period due to the billion-year half-lives of the heat-producing elements responsible for geothermal heat” [67-68] – how important is this mechanism in producing subglacial liquid water without tidal heating (e.g. in planets too far from their hosts)? And vice versa, can tidal heating maintain basal melting on planets that do not have enough geothermal radioactive heat production? ([159-162]).

We thank the reviewer for their constructive comments. The primary role of this paper is to compute the heat flow required for basal melting and to assess if geothermal energy can be sufficient, but you are absolutely correct in stating that tidal energy could be another notable source of energy. This may be especially true for old exo-Earths such as the ones in the TRAPPIST 1 system. We have significantly added some discussion of this topic added the following paragraph to address this point:

“While reasonable constraints on T_s of exo-Earths can be placed based on the estimated T_{eq} , heat flow on exo-Earths is entirely unconstrained. The main source of long-term heat in a planet after the initial stage of accretion and differentiation is the radiogenic decay of isotopes of heat-producing elements with billion-year half-lives such as ^{40}K , ^{232}Th , ^{235}U , and ^{238}U (i.e., geothermal heat). For example, $\sim 80\%$ of the Earth’s present-day surface heat flow can be attributed to the decay of radioactive isotopes [50]. Radiogenic heat production as a function of age for cosmochemically Earth-like exoplanets suggests that exo-Earths similar in age to Earth should have a comparable heat production rate (H) [51]. However, there is a considerable degree of variations and uncertainties associated with the age of the M-dwarf systems considered in this study (Table. 1). A comparison of the H required for basal melting on the various exo-Earths to the age-dependent heat production values of cosmochemically Earth-like exoplanets (e.g., [51]) is shown in Figure 9. Despite the old age of some of the M-dwarf systems, the heat production rate on these exo-Earths may be sufficient for basal melting if they are cosmochemically Earth-like. The notable exceptions are TRAPPIST-1 f and TRAPPIST-1 g, where basal melting of thin ice sheets by geothermal heat alone may not be feasible given their old age (hence lower radiogenic heat production) and their relatively low surface temperature (also see Supplementary Figure 2).

[13] A comparison of the heat flow required for basal melting on various exo-Earths to the heat flow on moons and planets of our solar system may also allow us to further contextualize whether basal melting would be feasible on exo-Earths. The mean oceanic and continental heat flow on Earth is 101 mW.m^{-2} and 65 mW.m^{-2} , respectively (Fig. 4 (e)); however, in active hot spots like Yellowstone and mid-oceanic ridges, the surface heat flow can exceed 1000 mW m^{-2} [52]. On the Moon, heat-flow was measured to be $21 \pm 3 \text{ mW.m}^{-2}$ and $15 \pm 2 \text{ mW.m}^{-2}$ at Apollo 15 and 17 landing sites respectively [53]. No direct measurements of the Martian surface heat flow are currently available; however, indirect remote sensing and models indicate heat flow up to 25 mW.m^{-2} [54]. A first-order comparison of the heat flow required for basal melting on various exo-Earths to the heat flow of Earth provides further corroboration that basal melting may be entirely feasible on most exo-Earths with Earth like WMF (Fig. 9). For thick ($>3 \text{ km}$) ice sheets, the heat production rate on exo-Earths can be $\sim 50\%$ that of Earth’s, and basal melting can still occur (Supplementary Figure. 2). The heat production rate on Kepler 442 b can be $\sim 5\%$ that of Earth’s, and basal melting could still occur. The thin dotted lines in Fig. 9 (e) show the range of heat flow for Moon and Mars. In some planets like

Kepler 442 b, Proxima Centauri B, and TRAPPIST-1e, even ice sheets that are 1-2 km thick may undergo basal melting with Mars-and-Moon like heat flow (Supplementary Figure 2). Heat flows in excess of Earth's continental and oceanic average are only required to melt thin ice sheets on planets with extremely low T, such as Trappist-1 g, GJ 667 C e, and Kepler-186 f (Fig. 4; Fig. 6; Supplementary Figure. 2). It is conceivable that the exo-Earths considered in this study are not cosmochemically Earth like and the geothermal heat flow on those exo-Earths may not be sufficient for basal melting. In such a scenario, tidal heating may provide additional source of the heat on some exo-Earths around the habitable zone of M-dwarf stars [55]. For example, the age of the TRAPPIST-1 system is estimated to be 7.6 ± 2.2 Gyr [56]; thus, if geothermal heating has waned more than predicted by the age-dependent heat production rate assumed here [51], tidal heating could be an additional source of heat for basal melting on the TRAPPIST-1 system [57, 58]."

Running comments:

62. before "devoid of adequate greenhouse warming" - add "outside the Habitable Zone and".

Thanks for the suggestion. We have fixed this sentence.

75. [1] (Korappapu 2013) - it would be nice to add a more recent ref, also relating to sunlike planets, eg. Bryson et al 2021 AJ 161 36.

We added a more recent work that reports new potentially habitable planets orbiting M dwarfs from the full Kepler data set.

(Dressing, C. D., & Charbonneau, D. (2015). The occurrence of potentially habitable planets orbiting M dwarfs estimated from the full Kepler dataset and an empirical measurement of the detection sensitivity. *The Astrophysical Journal*, 807(1), 45.)

76-77. lifetimes of hundreds of billions of years - this is a common yet meaningless argument, lifetime (like our sun) is quite enough and M dwarf has no advantage in this respect.

We stated the importance of prolonged life of M-dwarfs system based on the theoretical work by Loeb et al. (2016), in which the authors used a probabilistic model to look at the relative likelihood for the emergence of life as a function of cosmic time. In that work, the authors found that for low mass stars, the peak probability for emergence of life occurs much later in future. The primary line of reasoning is that M dwarfs have an extended pre-main sequence (PMS) phase during the first billion years or so of their life time. This PMS phase is characterized by saturated emission of X-ray and UV. By the time, M dwarf has settled onto the Main Sequence, planets that were once habitable may have lost oceans worth of water to space, and could be long desiccated and void of surface life. That is not to say, that the planets could not have retained or regained water in later years; thus, all else being equal given the evolutionary phase of M dwarf, the prolonged life scale may offer an edge over other massive stars. However, we have incorporated your suggested clarification that high energy radiation only affects the upper layers of water.

Reference:

*Loeb, A., Batista, R. A., & Sloan, D. (2016). Relative likelihood for life as a function of cosmic time. *Journal of Cosmology and Astroparticle Physics*, 2016(08), 040.*

82- 84. "X-rays and UV ... more flaring...presents challenges for the surface habitability of these planets"

- this too is not a very strong reservation, as for aquatic life 20 cm of water can effectively shield these radiations, and water may be retained or regained by M-dwarf planets in spite of the early violent evolutionary epoch of their host, eg. Gale and Wandel 2017 IJA 16 1.

We have revised the introductory paragraph and have softened the tone regarding the effects X-ray, UV and flaring could pose to the surface habitability by stating:

“may present some challenges for the surface habitability of these planets.”

We have also added the following line:

“Although such radiation only penetrates ~20 cm of liquid water [12], this radiation still poses a challenge for near-surface aquatic environments where ocean circulation will periodically expose water from below this depth to the surface.”

94. “basal melting may provide an alternative means of forming liquid water in a subsurface environment shielded from high-energy radiation (Fig. 1)” - According to Fig (1), a thick intraglacial ocean is present between the Ih and V layers. It is also well protected from energetic radiation. It should at least be motivated, why this is not enough for habitability and what is the added contribution of a basal melting layer (e.g. could the main ocean exist without a basal melting layer?)

Indeed, an intraglacial ocean by itself could be great for habitability. However, these intraglacial ocean may only be present on planets that are substantially water-rich, possibly planets like LHS 1140 B. We have also added the following lines in the paper to discuss the habitability potential of the intraglacial ocean.

“An isolated interglacial ocean may also form, without chemical exchange (i.e. nutrient flux) with the rocky core, which may not be adequate for habitability [60]. However, liquid aqueous solutions generated by basal melting at the rock/high-pressure ice mantle boundary will be buoyant and will permit to dissolve and transport important life sustaining solutes/nutrients from the rocky core to the interglacial ocean.”

127. WMF should be defined here rather than in [132]

Thanks. Fixed now.

154. If the masses are similar to Earth (or 6 times larger in the case of 1140b) why are only values up to 60 mW/m² considered? In [148] it is stated that Earth's heat flow is 60-100 mW/m².

Figure 4 now shows the range of heat flow possible on Earth. While it is true that heat flow on Earth can even exceed 1000 mW m⁻² near hotspots and mid oceanic ridges, we wanted a conservative estimate on the feasibility of basal melting. We have removed the line where we say only values up to 60 mW m⁻² are considered. Instead, we now report heat flow values required for basal melting as a function of various parameters and comment on the likeliness of those heat flow values being present on the various exo-Earths considered here. We use modeled radiogenic heat production rate for cosmochemically Earth like planets to further assess the feasibility of basal melting and if the exo-Earths under consideration can have such heat flow.

204. ex Earths

Thanks. Fixed now.

216. Fig.3 – Heat production rate is monotonously decreasing with thickness in the range of 1-10 km. How is this consistent with Fig. 1, which indicates a non-monotonous temperature dependence of the ice-liquid boundary in this range?

We are unclear on the exact definition of ‘monotonous’ in this scenario. We assume that the reviewer is saying non-linear temperature dependence? Figure 1 (now Figure 2 in this version) shows the depth-melting point relationship on the super Earth LHS 1140 b that has a surface gravity of 23 m s^{-2} . The heat production rate we show in Supplementary Figure 2 of this version is for exo-Earths with Earth like mass and radius. The basal pressure at the ice-crust interface never exceeds the threshold required for the existence of high-pressure ice polymorphs. Thus, heat production rate monotonously decreases as the melting point of Ice-Ih also linearly decreases with pressure.

218. GEL is undefined in the main ms (only in the Supp. File [168])

Thanks. Fixed now.

237. Supp/ Fig 7 shows that actually there is hardly any evolution

We have revised this figure (now figure S3) and the evolution is much more apparent now.

239 “due to the pressurized ice at the base, which has a reduced melting point (Fig. 1).” – how is this shown in Fig 1? Same question for [249].

The reduction in melting point with increasing pressure for ice phase Ih is shown in Figure 1b. The increase in melting temperature with ice thickness is also shown in Figure 1b. For example in the figure below, the red sketch shows how the melting temperature of ice initially decreases with pressure.

324. What happens after the 5-million-year limit? This is a short time compared with the evolution of life. The author needs to comment on this. However, Fig. 2 indicates a steady state is reached after less than 2 million years for Proxima Cen b. Is this typical also for the other planets and for different parameter values? Assuming a steady state and neglecting secular cooling may underestimate the heat input required in order to keep liquid water for extended biological periods (billions of years).

We have added the following text to address this comment:

“For computational efficiency and to demonstrate the feasibility of basal melting, we have limited the thermophysical evolution of ice for 5 million years. This time frame is adequate for the thermal equilibration of ice sheets and to ascertain whether a given heat flow is sufficient for basal melting. To assess the stability of basal melt over timescales relevant to the genesis of life on Earth (0.5 – 1 Gyr), we model the thermophysical evolution of a 1-km ice sheet on Proxima Centauri B for a billion years (Fig. 8). Over a billion years, geothermal heat flow can be expected to decline given the billion years half-life of most heat producing elements. Thus, we consider two scenarios: an extreme heat loss scenario in which the heat flow on Proxima Centauri B exponentially declines from the current Earth-like heat flow of 60 mW m^{-2} to the current Mars-like heat flow of 30 mW m^{-2} within a billion years. In this rapid heat loss scenario basal melt would only be stable for ~ 750 million years. In the second, moderate heat loss scenario, the same amount of heat flow declines over a 4-billion-year period. As shown in Figure 8, basal melt in an exo-Earth with moderate heat loss can persist for a geologically prolonged period (>3 Gyrs). Supplementary Figure. 6 shows similar results for the coldest exo-Earth in our study, TRAPPIST-1 g. The efficiency of heat loss on exo-Earths and its impact on basal melting is further discussed in section 3.”

337. Give ref or evaluate eq 2

We provide a reference to Eq 2.

339. c should be cP.

Thanks. Fixed now.

341. "...thermal conductivity are iteratively calculated using Sea-Freeze or Table 1" - how does this comply with the above statement [313] that Sea-Freeze does not model variations in thermal conductivity?

We meant to say that density and specific heat are iteratively calculated using Sea-Freeze, but the thermal conductivity is based on the values provided in Table-1. We have revised this sentence to avoid this confusion and now it states:

"The pressure and temperature dependence of density and specific heat are iteratively calculated using Sea-Freeze. The ice-phase dependency of thermal conductivity is updated based on the values reported in Table 1."

350. Give ref or evaluate eq. 3

We provide a reference to Eq 3.

380. Apparently eq. 6 gives values only slightly different from eq. 2 unless R_a is very different from R_{ac} . What is the expected range of R_a ? What is typically the difference between the model including solid-state convection and calculating using only eq. 2, for all regions?

The main difference introduced by solid-state convection is that the thickness of the basal melt decreases, which is to be expected as convection should amplify heat loss efficiency through the shell. This is why we think this model with solid-state convection is more realistic. We provide an example below for Proxima Centuari B. The examples below show the thermophysical evolution of a 10 km ice sheet with (right) and without convection (left). As can be seen, the thickness of the basal melt is notably thinner when we consider convection.

The expected range of Ra depends on the particular model. For the case above: in the final temperature profile below, you can see an adiabatic profile in the water layer and the convecting portion of the ice shell, and you can see that the new boundary layer thicknesses are different at the top and the bottom of the convecting layer. The final plot shows the Rayleigh number as a function of depth and time. We are not sure that plots like these would be helpful to the larger audience, so we do not include these plots in the paper.

395-396. Give ref. or define/explain.

We provide reference to two previous papers where this methodology is described in detail.

402. Define c_j (cP)?

C_j is the specific heat of the j th layer and we have revised line 402 as follows:

“where H_j and c_j are the enthalpy and specific heat of the j th discretized layer.”

425. The metallic core of a planet cannot contain a significant fraction of the heat-producing element - why? also, should be plural (elements).

This is due to the lithophilic nature of the heat producing elements like K, Th, and U. We have revised this sentence as follows:

“Due to the lithophilic nature of heat-producing elements, they are assumed to be absent in the metallic core of a planet;”

REVIEWER COMMENTS

Reviewer #1 (Remarks to the Author):

Overall I thank the authors for the revisions they have made following the three reviews which have addressed many of the queries raised by the reviewers. The two main points I raised, concerning the number of experiments and overall ordering of the paper have been addressed well. I like the new figures 4 and 6, placing the various exo-earths into the overall parameter space investigated, and the numbered sections, and expansion of the introduction and other sections certainly improves the flow and overall clarity of the paper, and answers my main comments, and many of the specific ones as well. I am happy for it to proceed to publication, though I have some small additional comments/suggestions where I think the clarity could be improved, or where suggested changes do not seem to have reached this version.

In particular, paragraph 2 does not seem to have changed; the tracked change version shows no changes, so I assume that the authors have inadvertently not included the changes suggested, and which they refer to as being made in their rebuttal. The original reference to west Antarctic ice sheet lakes at line 109 is still there, and the recent Livingstone paper I suggested has not been incorporated. At around line 116, the Lauro et al. paper I suggested again has not been incorporated. There have also been several rather recent papers which have further addressed the possibility of basal melting beneath Mars' south polar ice cap:

Lalich, D.E., Hayes, A.G. & Poggiali, V. Explaining Bright Radar Reflections Below The South Pole of Mars Without Liquid Water. *Nat Astron* 6, 1142–1146 (2022). <https://doi.org/10.1038/s41550-022-01775-z> (replaces current reference 28)

Arnold, N.S., Butcher, F.E.G., Conway, S.J. et al. Surface topographic impact of subglacial water beneath the south polar ice cap of Mars. *Nat Astron* (2022). <https://doi.org/10.1038/s41550-022-01782-0>

Lauro, S.E., Pettinelli, E., Caprarello, G. et al. Using MARSIS signal attenuation to assess the presence of South Polar Layered Deposit subglacial brines. *Nat Commun* 13, 5686 (2022). <https://doi.org/10.1038/s41467-022-33389-4>

I realise this debate is not central to the arguments within the paper, but it is still live!

Line 177 – change ‘prevalent’ to ‘common’ or ‘widespread’; ‘prevalent’ can suggest ‘in the majority’ in some situations, which seems to me a little strong in this case.

Line 233 – ‘previous studies...ice mantles...’. For this paragraph, I also wonder if Supplementary Figure 3 could be combined with Figure 7. Supplementary Figure 3 could become Fig 7 a-d, with the current Figure 7 as 7e ‘underneath’ (akin to Figure 9, say).

Line 331 – delete room (room pressures)

Line 338 – change ‘in Europa’ to ‘on Europa’; just reads more naturally to me.

Line 350 – I betray myself as a terrestrial glaciologist here, but ‘interglacial’ has a specific meaning of ‘warm period within the Quaternary’ and I think this term should be avoided here. I know what the authors are trying to say, but how about something like ‘ice-interstitial ocean’? Or explain the situation more fully – ‘An isolated ocean between layers of low density, low pressure ice and high density, high pressure ice...’

References

As well as the omissions (and potential new references) discussed above, several references are incomplete, e.g. 13, 14, 15, 16, 17, 18 (should be updated to the accepted version: Rutishauser, A., Blankenship, D. D., Young, D. A., Wolfenbarger, N. S., Beem, L. H., Skidmore, M. L., Dubnick, A., and Criscitiello, A. S.: Radar sounding survey over Devon Ice Cap indicates the potential for a diverse hypersaline subglacial hydrological environment, *The Cryosphere*, 16, 379–395, <https://doi.org/10.5194/tc-16-379-2022>, 2022.) 19, 23, 24, 25, 27, 30, 31, 53, 64.

Reviewer #2 (Remarks to the Author):

Review of revised manuscript NCOMMS-22-24296A: Liquid Water on Cold Exo-Earths via Basal Melting of Ice Sheets, by Ojha et al.

Reviewed by S. Clifford.

Summary comments:

The authors identify the minimum thermal and pressure conditions under which liquid water might persist beneath ice covers of varying thicknesses for 12 examples of exo-Earths. The expanded discussion in the revision has addressed all of the points I identified in my original review. The findings are important and original and indicate that for planets with thick hydrospheres, there may exist a potentially complex stratigraphy of habitable liquid water environments at depth. I believe that the manuscript, as revised, warrants publication in Nature Communications.

Reviewer #3 (Remarks to the Author):

Thanks for the detailed response.

On the questions about geothermal heating, the authors added the sentence " For example, the age of the TRAPPIST1 system is estimated to be 7.6 ± 2.2 Gyr [56]; thus, if geothermal heating has waned more than predicted by the age-dependent heat production rate assumed here [51], tidal heating could be an additional source of heat for basal melting on the TRAPPIST-1 system [57, 58]."

I expected to see an estimate of the geothermal heating, particularly since in the case of this example, the Trappist-1 planets which are the furthest ones and hence may need additional heating (g and h) also have the lowest rate of geothermal heating.

76-77. lifetimes of hundreds of billions of years - this is a common yet meaningless argument, lifetime (like our sun) is quite enough and M dwarf has no advantage in this respect.

Perhaps I did not make myself clear, so the authors have misunderstood this comment. I meant to say simply that lifetimes of hundreds of billions of years are not more advantageous than 10 billion years, given the age of the universe.

216. Fig.3 – Heat production rate is monotonously decreasing with thickness in the range of 1-10 km. How is this consistent with Fig. 1, which indicates a non-monotonous temperature dependence of the ice-liquid boundary in this range?

Fig. 3 is now 3b, Fig. 1 is now fig 2b. I meant the decrease of the temperature with depth on the boundary between the liquid-solid in fig.1 (now 2).

KEY

BOLD: Reviewer's Query

Italicized: Response to the Reviewers

Italicized & Underlined: Revised Text from the New Version

Reviewer #1 (Remarks to the Author):

Overall I thank the authors for the revisions they have made following the three reviews which have addressed many of the queries raised by the reviewers. The two main points I raised, concerning the number of experiments and overall ordering of the paper have been addressed well. I like the new figures 4 and 6, placing the various exo-earths into the overall parameter space investigated, and the numbered sections, and expansion of the introduction and other sections certainly improves the flow and overall clarity of the paper, and answers my main comments, and many of the specific ones as well. I am happy for it to proceed to publication, though I have some small additional comments/suggestions where I think the clarity could be improved, or where suggested changes do not seem to have reached this version.

Dear Reviewer,

We thank you for your constructive, detailed, and prompt review (one more time). We have made the small additional changes as you suggested.

In particular, paragraph 2 does not seem to have changed; the tracked change version shows no changes, so I assume that the authors have inadvertently not included the changes suggested, and which they refer to as being made in their rebuttal. The original reference to west Antarctic ice sheet lakes at line 109 is still there, and the recent Livingstone paper I suggested has not been incorporated.

With all the different versions of manuscripts circulated amongst the co-authors, somehow, the references you mentioned in the first review got omitted. We have added the missing references and made the other changes you suggested. We refer to the Livingstone paper as a general paper relevant to the distribution of subglacial lakes and retain the old references for subglacial lakes in specific regions of Earth.

“Basal melting is responsible for the formation of subglacial liquid water lakes in various areas of Earth [13], such as the West Antarctic Ice Sheet [14], Greenland [15-17], and possibly the Canadian Arctic [18, 19].”

At around line 116, the Lauro et al. paper I suggested again has not been incorporated. There have also been several rather recent papers which have further addressed the possibility of basal melting beneath Mars' south polar ice cap:

– Lalich, D.E., Hayes, A.G. & Poggiali, V. Explaining Bright Radar Reflections Below The South Pole of Mars Without Liquid Water. *Nat Astron* 6, 1142-1146 (2022). <https://doi.org/10.1038/s41550-022-01775-z> (replaces current reference 28)

– Arnold, N.S., Butcher, F.E.G., Conway, S.J. et al. Surface topographic impact of subglacial water

beneath the south polar ice cap of Mars. Nat Astron (2022). <https://doi.org/10.1038/s41550-022-01782-0>

– Lauro, S.E., Pettinelli, E., Caprarelli, G. et al. Using MARSIS signal attenuation to assess the presence of South Polar Layered Deposit subglacial brines. Nat Commun 13, 5686 (2022). <https://doi.org/10.1038/s41467-022-33389-4>

I realise this debate is not central to the arguments within the paper, but it is still live!

We have added the Lauro et al paper and the other recent papers in this section.

“Although much debated (e.g., [28-31]), basal melting may also be responsible for the formation of a putative subglacial lake in the south pole of Mars today [32-36], where the mean annual surface temperature lies around 165 K.”

Line 177 – change ‘prevalent’ to ‘common’ or ‘widespread’; ‘prevalent’ can suggest ‘in the majority’ in some situations, which seems to me a little strong in this case.

We agree and have replaced the word ‘prevalent’ with ‘common’.

“Our findings suggest that exo-Earths resembling the snowball Earth or the icy moons of Jupiter and Saturn may be common in the Milky-way galaxy.”

Line 233 – ‘previous studies...ice mantles...’. For this paragraph, I also wonder if Supplementary Figure 3 could be combined with Figure 7. Supplementary Figure 3 could become Fig 7 a-d, with the current Figure 7 as 7e ‘underneath’ (akin to Figure 9, say).

Thank you for this suggestion. We have combined Supplementary Figure 3 with Figure 7 and revised the texts accordingly.

Line 331 – delete room (room pressures)

Thanks. Done.

Line 338 – change ‘in Europa’ to ‘on Europa’; just reads more naturally to me.

Thanks. Done.

Line 350 – I betray myself as a terrestrial glaciologist here, but ‘interglacial’ has a specific meaning of ‘warm period within the Quaternary’ and I think this term should be avoided here. I know what the authors are trying to say, but how about something like ‘ice-interstitial ocean’? Or explain the situation more fully – ‘An isolated ocean between layers of low density, low pressure ice and high density, high pressure ice...’

Thanks. Done.

References

As well as the omissions (and potential new references) discussed above, several references are incomplete, e.g. 13, 14, 15, 16, 17, 18 (should be updated to the accepted version: Rutishauser, A., Blankenship, D. D., Young, D. A., Wolfenbarger, N. S., Beem, L. H., Skidmore, M. L., Dubnick, A., and Criscitiello, A. S.: Radar sounding survey over Devon Ice Cap indicates the potential for a diverse hypersaline subglacial hydrological environment, *The Cryosphere*, 16, 379–395, <https://doi.org/10.5194/tc-16-379-2022>, 2022.)

19, 23, 24, 25, 27, 30, 31, 53, 64.

We use 'EndNote' software as our references manager and it looks like it made some error in reference list. Thank you for noticing these errors. We have manually updated the references.

Reviewer #2 (Remarks to the Author):

Review of revised manuscript NCOMMS-22-24296A: Liquid Water on Cold Exo-Earths via Basal Melting of Ice Sheets, by Ojha et al.

Reviewed by S. Clifford.

Summary comments:

The authors identify the minimum thermal and pressure conditions under which liquid water might persist beneath ice covers of varying thicknesses for 12 examples of exo-Earths. The expanded discussion in the revision has addressed all of the points I identified in my original review. The findings are important and original and indicate that for planets with thick hydrospheres, there may exist a potentially complex stratigraphy of habitable liquid water environments at depth. I believe that the manuscript, as revised, warrants publication in Nature Communications.

We thank Dr. Clifford for their review.

Reviewer #3 (Remarks to the Author):

Thanks for the detailed response.

On the questions about geothermal heating, the authors added the sentence " For example, the age

of the TRAPPIST1 system is estimated to be 7.6 ± 2.2 Gyr [56]; thus, if geothermal heating has waned more than predicted by the age-dependent heat production rate assumed here [51], tidal heating could be an additional source of heat for basal melting on the TRAPPIST-1 system [57, 58].”

I expected to see an estimate of the geothermal heating, particularly since in the case of this example, the Trappist-1 planets which are the furthest ones and hence may need additional heating (g and h) also have the lowest rate of geothermal heating.

We understand your point now. Thank you for the additional clarification. While some estimate of the geothermal heat production is possible, for example, using the approach here, it is much harder to estimate the geothermal heat flow. This is primarily because geothermal heat flow also depends on the mode of the heat transfer. For example, on Earth, divergent boundaries have the highest surface heat flow because of the direct nature of heat transfer that also creates new crust from the molted mantle material. However, if the geothermal heat production were to remain the same but the mode of heat transfer was in a thick lithosphere via conduction only, the surface heat flow may be lower. Without a sense of what the heat transfer mechanism may look like on these exo-Earths, any estimation of heat flow from heat production may be riddled with uncertainty. In essence, what we show here is that the old, cosmochemically Earth-like exo-Earths at least have the adequate heat budget to provide the required heat flow for basal melting.

76-77. lifetimes of hundreds of billions of years - this is a common yet meaningless argument, lifetime (like our sun) is quite enough and M dwarf has no advantage in this respect. Perhaps I did not make myself clear, so the authors have misunderstood this comment. I meant to say simply that lifetimes of hundreds of billions of years are not more advantageous than 10 billion years, given the age of the universe.

Thanks for the clarification. We understand your point now much better now and have removed this statement.

216. Fig.3 - Heat production rate is monotonously decreasing with thickness in the range of 1-10 km. How is this consistent with Fig. 1, which indicates a non-monotonous temperature dependence of the ice-liquid boundary in this range?

Fig. 3 is now 3b, Fig. 1 is now fig 2b. I meant the decrease of the temperature with depth on the boundary between the liquid-solid in fig.1 (now 2).

We apologize if we misunderstood your comment once again here and we are more than happy to provide more elaboration.

Although it may look like there is a non-monotonous temperature dependence of ice-liquid boundary (i.e., melting temperature of water-ice) in the 1-10 km range in Figure. 2, the melting temperature is indeed decreasing in a monotonic fashion.

Mathematically, a function is classified as monotonically decreasing if $f(x_1) > f(x_2)$ between x_1 and x_2 . Thus, if $\Delta f(x) \leq 0$ throughout the range of $\{x_1, x_2\}$, a function can be classified as monotonically decreasing function (ref).

At pressures below ~ 200 MPa (or in the range of 1-20 km thick ice sheets on Proxima Centauri b), the temperature dependence of ice-liquid boundary (i.e., the melting temperature) is indeed monotonously decreasing, as shown in the figure below. The left Y-axis shows the melting temperature as a function of depth (assuming surface gravity of Proxima Centauri b) and the right Y-axis shows the gradient of the melting temperature. The bottom plot shows the same relation but as a function of Pressure in megapascals.

At each spatial discretized level, $\Delta f(x) \leq 0$, thus melting temperature is decreasing monotonic fashion. The heat flow required for basal melting thus would also decrease in a monotonic fashion.

REVIEWER COMMENTS

Reviewer #1 (Remarks to the Author):

Thank you to the authors for making the suggested changes. I am happy for the article to move forward to publication.

Reviewer #3 (Remarks to the Author):

The authors have responded to the remaining 3 issues, but on two I have probably still not made myself clear:

1) Estimates of geothermal heat - I meant including tidal heating. The discussion [300-307] is now clear, but a simple estimate of the tidal heating rate, e.g. of Trappist-1g could improve the presentation.

2) By "non-monotonous" I referred to the phase diagram (fig 2b). of ice. The authors responded:

"Although it may look like there is a non-monotonous temperature dependence of ice-liquid boundary (i.e., melting temperature of water-ice) in the 1-10 km range in Figure. 2, the melting temperature is indeed decreasing in a monotonic fashion."

Perhaps I misunderstand something, but to me, the temperature of the water-ice (Ih) boundary for a depth between 0-3km is clearly decreasing with depth (and pressure), while for larger depths (or pressures) the temperature of the water-ice(III, V, VI) boundary is increasing with depth (and pressure), which apparently contradicts with figs. 3a and 3b, where the melting temperature is increasing with depth and pressure between 0-2 km. I think this (perhaps only apparent) contradiction should be clarified in the text.

Response to Reviewers – NCOMMS-22-24296-T

KEY

BOLD: Reviewer's Query

Italicized: Response to the Reviewers

Italicized & Underlined: Revised Text from the New Version

Reviewer #1 (Remarks to the Author):

Thank you to the authors for making the suggested changes. I am happy for the article to move forward to publication.

We thank Reviewer #1 for the review.

Reviewer #3 (Remarks to the Author):

The authors have responded to the remaining 3 issues, but on two I have probably still not made myself clear:

1) Estimates of geothermal heat - I meant including tidal heating. The discussion [300-307] is now clear, but a simple estimate of the tidal heating rate, e.g. of Trappist-1g could improve the presentation.

In the traditional Earth-centric geological context (or, maybe just in my head), geothermal heat implies heat primarily derived from planet's interior via the initial accretion, differentiation, and radiogenic decay of heat producing elements, and does not consider any possible tidal heating. Therefore, we were confused with this comment.

The situation with M-dwarf orbiting exo-Earths is quite different and the total heat flow can have significant tidal heating component, so we understand your comment now. We have added the following lines regarding the possibility/estimates of tidal heating on TRAPPIST-1 system:

“On planets e and f of the TRAPPIST-1 system, tidal heating is estimated to contribute heat flow between 160 and 180 mW m⁻² [60]. Thus, even if geothermal heating were to be negligible on these bodies, basal melting could still occur via tidal heating alone (Supplementary Figure. 2). However, for TRAPPIST-1 g, the mean tidal heat flow estimate from N-body simulation is less than 90 mW m⁻² [43]. Thus, ice sheets thinner than a few kilometers are unlikely to undergo basal melting on TRAPPIST-1 g (Supplementary Figure. 2).”

Two other exo-Earth where tidal heating may play a notable role in basal melting are Kepler 186 f and GJ 667 C e, however, we could not find any estimates of tidal heating for these bodies.

2) By "non-monotonous" I referred to the phase diagram (fig 2b). of ice. The authors responded: "Although it may look like there is a non-monotonous temperature dependence of ice-liquid boundary (i.e., melting temperature of water-ice) in the 1-10 km range in Figure. 2, the melting temperature is indeed decreasing in a monotonic fashion."

Perhaps I misunderstand something, but to me, the temperature of the water-ice (II) boundary for a depth between 0-3km is clearly decreasing with depth (and pressure), while for larger depths (or pressures) the temperature of the water-ice(III, V, VI) boundary is increasing with depth (and pressure), which apparently contradicts with figs. 3a and 3b, where the melting temperature is increasing with depth and pressure between 0-2 km. I think this (perhaps only apparent) contradiction should be clarified in the text.

We understand your point now. The following equation gives the depth-pressure dependence as shown in Figure 2 (b):

$$P(z) = \rho(z)gz;$$

where P and ρ are pressure and density as a function of depth (z) and g is the surface gravity. Figure 2(b) is simply showing an example of the pressure-depth relationship using the above equation for the super-Earth LHS 1140 b, which has a high surface gravity of 23 m s⁻². Thus, the right Y-axis on

that figure is not generally applicable to other exo-Earths that have different surface gravity values. We do mention this in the caption of Figure. 2.

“The right vertical scale shows an example of the approximate depth scale for an 80 km ice sheet on exo-Earth LHS 1140 b, assuming a surface gravity of 23 m. s^{-2} .”

Below, we show the dependence of melting temperature on the depth-pressure relationship for four different planets with very different surface gravity (both the names of the planets and their corresponding surface gravity are in the title of the plots). As you can see, on planets with Earth-like mass, such as TRAPPIST-1 e, Earth, and GJ 667 C e, the melting temperature decreases monotonically with depth in the 1-20 km range.

You are absolutely right that on planets like LHS 1140 b, (bottom right figure), the melting temperature varies non-monotonically within the same depth range because the large surface gravity

of LHS 1140 b which results in the presence of high-pressure ice at shallow depths. However, the heat flow values required for basal melting, as shown in Supplementary Figure 2, are for exo-Earths with Earth-like mass only (i.e., results from LHS 1140 b are not included). The heat flow required for basal melting for super-Earth like LHS 1140 B is shown in Figure 7 and you can see the non-monotonous relationship between pressure-melting temperature-heat flow in that figure (but do also keep in mind that the simulations were run at 5-km spacings in this figure, so not every kink is visible). This is also why we separate the results section for exo-Earths with masses like Earth and super-Earths.

To avoid this confusion, we have replaced Figure 2 with a new version (shown below) where we have removed the depth range on the right-Y-axis.

We have also removed the following line from Figure 2 caption:

“The right vertical scale shows an example of the approximate depth scale for an 80 km ice sheet on exo-Earth LHS 1140 b, assuming a surface gravity of 23 m. s^{-2} .”

We hope this provides additional clarification.

REVIEWERS' COMMENTS

Reviewer #3 (Remarks to the Author):

Thanks, the authors have now satisfactorily replied to my comments and consequently, I believe, improved the ms. I hence approve publication.

Response to Reviewers – NCOMMS-22-24296-T

KEY

BOLD: Reviewer's Query

Italicized: Response to the Reviewers

Italicized & Underlined: Revised Text from the New Version

Reviewer #3 (Remarks to the Author):

Thanks, the authors have now satisfactorily replied to my comments and consequently, I believe, improved the ms. I hence approve publication.

We would like to thank the reviewer again for their thorough review.